# Peptidoglycan biosynthesis is driven by lipid transfer along enzyme-substrate affinity gradients

Abraham O. Oluwole [1,2], Robin A. Corey[3], Chelsea M. Brown[4], Victor M. Hernández-Rocamora[5], Phillip J. Stansfeld [3,4], Waldemar Vollmer[5], Jani R. Bolla [2,6✉] & Carol V. Robinson [1,2✉]

Maintenance of bacterial cell shape and resistance to osmotic stress by the peptidoglycan (PG) renders PG biosynthetic enzymes and precursors attractive targets for combating bacterial infections. Here, by applying native mass spectrometry, we elucidate the effects of lipid substrates on the PG membrane enzymes MraY, MurG, and MurJ. We show that dimerization of MraY is coupled with binding of the carrier lipid substrate undecaprenyl phosphate ($C_{55}$-P). Further, we demonstrate the use of native MS for biosynthetic reaction monitoring and find that the passage of substrates and products is controlled by the relative binding affinities of the different membrane enzymes. Overall, we provide a molecular view of how PG membrane enzymes convey lipid precursors through favourable binding events and highlight possible opportunities for intervention.

[1] Physical and Theoretical Chemistry Laboratory, University of Oxford, South Parks Road, Oxford OX1 3QZ, UK. [2] The Kavli Institute for Nanoscience Discovery, South Parks Road, Oxford OX1 3QU, UK. [3] Department of Biochemistry, University of Oxford, South Parks Road, Oxford OX1 3QU, UK. [4] School of Life Sciences and Department of Chemistry, University of Warwick, Gibbet Hill Campus, Coventry CV4 7AL, UK. [5] Centre for Bacterial Cell Biology, Biosciences Institute, Newcastle University, Richardson Road, Newcastle upon Tyne NE2 4AX, UK. [6] Department of Plant Sciences/Biology, University of Oxford, Oxford OX1 3RB, UK. ✉email: jani.bolla@plants.ox.ac.uk; carol.robinson@chem.ox.ac.uk

The resurgence of antibiotic resistance is a serious threat to public health because of its grave clinical and economic impacts[1,2]. Since bacteria must preserve their cell envelope to avoid lysis and death, a key antibiotic target is peptidoglycan (PG), a net-like polymer of sugars and amino acids that provides bacteria with the means of maintaining their shape, rigidity, and tolerance to osmotic stress[3,4]. Accordingly, the use of antibiotics that compromise the biosynthesis of PG is one of the most successful strategies for combating bacterial infection[5,6]. For example, the penicillin-binding proteins (PBPs) and their substrates are the classical targets of antibiotics[7]. However, several pathogenic bacteria have developed a range of resistance mechanisms enabling them to escape from the potency of many conventional antibiotics[8,9]. Alternative targets such as the membrane-bound PG enzymes, therefore, represent promising but relatively underexplored drug targets[10,11]. Understanding the working mechanisms of existing antibiotics and the development of novel targets is critical to overcoming the problem of antibiotic resistance. This entails the deployment of novel strategies and methodologies to track core biosynthetic pathways with molecular-level precision as a prelude to testing antibiotic candidates.

PG is polymerised in the periplasm, meaning that the cytosolic precursor must be synthesised and transported in multiple steps across the cytoplasmic membrane. First, the integral membrane protein MraY catalyses the formation of lipid I from uridine 5′-diphospho-*N*-acetylmuramoyl-pentapeptide (UM5) and undecaprenyl phosphate ($C_{55}$-P)—the universal carrier lipid[12,13]. In the second step, the glycosyltransferase MurG catalyses the transfer of an *N*-acetylglucosamine residue from uridine 5′-diphospho-*N*-acetylglucosamine (UDP-GlcNAc) to lipid I[14]. The resulting product, lipid II, is then flipped to the periplasmic side of the inner membrane by MurJ[15,16] for subsequent incorporation into the PG polymer by PBPs and additional factors[4]. The structural architectures of MraY, MurG and MurJ, along with their interactions with substrates and inhibitors are known in some

detail[14,17–19]. However, how these proteins interact with the membrane-anchored ligands $C_{55}$-P, lipid I, and lipid II at the molecular level is often difficult to define using conventional structural approaches. For example, unresolved density profiles at the dimer interface of the MraY crystal structure can correspond to phospholipids or $C_{55}$-P[17]. Owing to the limitations of existing methodologies, the identity and structural roles of interfacial ligands/lipids is a subject of speculation.

Herein, we apply native mass spectrometry (MS) approaches combined with coarse-grained (CG) and atomistic molecular dynamic (MD) simulations to study the effects of lipid substrates on the PG biosynthetic membrane enzymes MraY, MurG and MurJ. Our data reveal a monomer-dimer equilibrium for MraY, and that binding of $C_{55}$-P at the interfacial sites mediates dimerisation. Further, we recapitulate the enzymatic activities of MraY and MurG to capture the effect of an antibiotic. By probing the relative affinities of MraY, MurG and MurJ for the peptidoglycan precursor lipids we find that, although MraY can bind its own reaction product lipid I, MurG and MurJ exhibit a strong preference for their respective natural substrates, lipid I and lipid II. Overall, we provide an in-depth molecular view of the core reactions in the biosynthesis of peptidoglycan precursors and present an approach for antibiotic screening.

## Results

**MraY is a substrate-mediated dimer.** To study the oligomeric state and activity of MraY by native MS, we purified the well-studied MraY homologue from *Aquifex aeolicus*[17,20,21]. When released into the mass spectrometer from a mild detergent (*n*-octyl-β-D-glucopyranoside (OG)), the resulting spectrum of MraY displays a series of charge states that correspond to monomers and dimers (Fig. 1a). Strikingly, a range of endogenous molecules copurified as noncovalent adducts with the dimer but not with the monomer. We assigned the most intense of these adducts to endogenous ligands: $C_{55}$-P (847.9 ± 1.0 Da) and

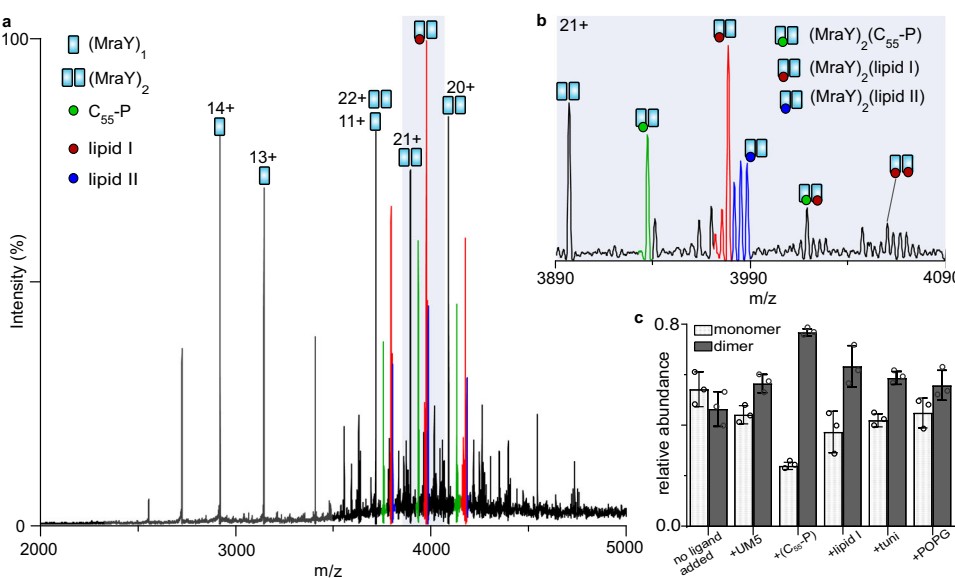

**Fig. 1 Endogenous ligand-mediated dimerisation of MraY. a** Native mass spectrum of 5 μM MraY liberated from 1% OG micelles. The charge states of MraY monomer and dimer are shown, with a large proportion of dimers binding endogenous $C_{55}$-P (847.9 ± 1.0) and lipid I (1716.7 ± 2.3 Da). Masses are given in Supplementary Table 1. **b** An expansion of the dimer charge state (21+) showing ligand-free and ligand-bound species. **c** Relative populations of monomer and dimer MraY upon incubation with a tenfold molar excess of UDP-MurNAc pentapeptide (L-Ala, D-Glu, L-Lys, D-Ala, D-Ala) (UM5), undecaprenyl phosphate ($C_{55}$-P), lipid I, tunicamycin (tuni.), and 1-palmitoyl-2-oleoyl-*sn*-glycero-3-phospho-(1′-rac-glycerol) (POPG). Full spectra are displayed in Fig. 3b and Supplementary Fig. 3. Data were presented as the mean (± SD), $n = 3$, while each circle represents an individual measurement value. Source data are provided as a Source data file.

lipid I ($1716.7 \pm 2.3$ Da). In addition, we observed low-intensity peaks that can be assigned to undecaprenyl diphosphate ($926.24 \pm 2.70$ Da), cardiolipin ($1402.90 \pm 1.83$ Da) and lipid II ($1920.09 \pm 2.11$ Da) copurifying with the MraY dimer (Fig. 1b and Supplementary Table 1). Consistent with these assignments, we performed tandem MS ($MS^2$) on ligand-bound MraY dimers and observed that the loss of these endogenous ligands results in ligand-free monomers as well as an apo MraY dimer that readily dissociates (Supplementary Fig. 1). This observation suggests that the bound ligands are providing structural stability to the MraY oligomer. In four additional detergents, we observed MraY monomers, dimers, and ligand-bound dimers (Supplementary Figs. 2, 3), which further indicates that the monomer-dimer equilibrium of MraY is mediated by ligands.

To examine the effects of MraY ligands on the observed monomer-dimer properties, we selected a C8E4-containing buffer as it enabled the removal of the majority of endogenous $C_{55}$-P and requires relatively low activation voltages to disrupt micelles[22,23]. Titration of MraY with exogenous $C_{55}$-P revealed that this ligand binds more favourably to MraY dimers than to the monomers under this condition (Fig. 1c and Supplementary Fig. 3). The binding of up to four $C_{55}$-P molecules per MraY dimer could be clearly resolved (Supplementary Fig. 3), indicative of binding interactions at multiple sites. Compared to the water-soluble substrate, UM5, the binding of $C_{55}$-P and lipid I significantly enhanced the relative proportion of dimeric MraY (Fig. 1c and Supplementary Fig. 3). Anionic phospholipids such as phosphatidylglycerol have been proposed to stabilise the MraY dimer[24]. We find however that the dimer-stabilising effect of $C_{55}$-P is considerably higher than that of 1-palmitoyl-2-oleoyl-*sn*-glycero-3-phospho-(1′-rac-glycerol) (POPG) under the same conditions (Fig. 1c and Supplementary Fig. 3). We also captured the binding of a nucleoside antibiotic tunicamycin to the MraY dimer. Tunicamycin binding also yielded an increase in the population of dimeric MraY (Fig. 1c and Supplementary Fig. 3). Although tunicamycin competes with the binding of UM5 at the catalytic site[25–27], our data suggest that the hydrophobic tails of tunicamycin can also stabilise the MraY dimer. Together these results suggest that MraY forms a more stable noncovalent complex with the lipid substrates as a dimer and that the lipid-like ligands are providing structural stability to the MraY oligomer.

To better understand the molecular nature of the interaction between MraY and its membrane-embedded substrates, we carried out extensive MD simulations of the MraY dimer (PDB ID: 5CKR) in a model lipid bilayer containing $C_{55}$-P, lipid I or both (see Methods). We analysed the interactions of MraY with $C_{55}$-P using a graph theory-based network modelling approach (https://github.com/wlsong/PyLipID)[28]. The data reveal the presence of several binding sites for $C_{55}$-P and lipid I around the MraY dimer which are occupied for at least 50% of the simulation time. These putative binding sites are seen on both periplasmic and cytoplasmic faces of the MraY dimer, but the most favourable interactions are predicted to be cytoplasmic, which are more likely to be physiologically relevant (Supplementary Figs. 4, 5). For $C_{55}$-P binding, the two sites with the highest predicted affinities (i.e., lowest $k_{off}$, see Methods) are a pair of equivalent sites at the dimer interface, on each side of the MraY protomers, with $k_{off}$ values of $\sim 1.0$ $\mu s^{-1}$ (Fig. 1e and Supplementary Fig. 4a). We also observe $C_{55}$-P interacting with MraY near the catalytic cleft and at other sites (Supplementary Fig. 4b–d), however the relatively faster kinetics ($k_{off} = 1.6 \pm 0.3$ $\mu s^{-1}$ and $1.5 \pm 0.1$ $\mu s^{-1}$) suggest that these interactions are weaker than at the dimer interface. The core of the interfacial binding sites of $C_{55}$-P is formed by Trp-253, Phe-254 and Gln-260 on one protomer and Leu-332, Lys-336 and Arg-340 on the other. One of

these residues, Arg-340, has the tightest binding from all residues in MraY, according to its $k_{off}$ value (Supplementary Table 3). Simulating $C_{55}$-P with an MraY protomer (see Methods) further indicates that $C_{55}$-P can bind to MraY via Arg-340, however, with appreciably lower affinity ($k_{off} = 2.1 \pm 0.3$ $\mu s^{-1}$) than when at the interfacial sites. When bound to the interfacial sites of the MraY dimer, a $C_{55}$-P molecule will have its phosphate group on the cytoplasmic face of the membrane and will bridge the two MraY protomers. In addition, the $C_{55}$-P tail will make extensive contact with both protomers.

When MraY was simulated with lipid I alone, we found that lipid I interact considerably at the interfacial and active site regions (Supplementary Fig. 5a, b). Lipid I also binds to other positions around MraY, albeit with lower affinity (Supplementary Fig. 5c, f). Interestingly, simulations of the MraY dimer with both $C_{55}$-P and lipid I, present in the model membrane, show that lipid I occupy the catalytic sites for a higher percentage of the simulation time than $C_{55}$-P (Supplementary Table 4). Analogously $C_{55}$-P occupies the interfacial sites for more of the simulation time than lipid I (Supplementary Table 4). Overall, these data indicate that MraY can interact with both $C_{55}$-P and lipid I at multiple sites, including around the dimer interface, and suggests a putative role of $C_{55}$-P and lipid I in conferring stability to the MraY dimer.

Guided by the interfacial $C_{55}$-P binding sites predicted by the MD simulation results, we generated several mutants of MraY to probe experimentally the link between substrate binding and dimerisation. Using identical expression and purification conditions employed for wild-type protein we examined K336A, Q260A, R340A, Q260A/R340A and W253A/F254A/R340A mutants of MraY. Native MS analyses showed that the point mutants Q260A and K336A MraY exhibited monomer and dimer properties similar to the wild type (Fig. 2b). However, dimers of both proteins were observed only in complex with endogenous lipid I, little or no protein-bound endogenous $C_{55}$-P was detected. For the Q260A mutant, each MraY dimer was detected in a complex with between 1–4 molecules of endogenous lipid I, indicating that the ligands can bind to MraY at other sites besides the canonical catalytic sites. Notably, the R340A mutation yielded a drastic reduction in the dimer population (Fig. 2a, b). Dimers bound to endogenous lipid I were observed but no ligand-free dimer. These observations indicated that residue R340 is critical to both dimerisation and binding of copurified $C_{55}$-P. Bearing in mind that $C_{55}$-P bound at the interfacial region is coordinated by equivalent residues on different protomers (Fig. 2c), we introduced an additional mutation at residue Q260. Interestingly, the double mutant Q260A/R340A yielded predominantly monomeric MraY, and a very small amount of lipid I-bound dimers (Fig. 2a, b). Similarly, the triple mutant W253A/F254A/R340A was also predominantly monomeric (Fig. 2a, b). Although R340 has the lowest $k_{off}$ from the MD simulation results, mutating this single residue was found to be insufficient to completely disrupt $C_{55}$-P binding and dimerisation. We attribute this observation to the extensive contacts made by the lipid substrate to both MraY protomers (Fig. 2c). However, a substantial disruption of the interfacial sites was achieved by mutating the complementary pair of residues on each protomer (Q260A/R340A and W253A/F254A/R340A). Consequently, MraY variants with these mutations are predominantly monomeric. Since the $C_{55}$-P coordinating residues at the interfacial sites may also be involved in stabilising the ligand-free form of the MraY dimer[17], there is a possibility that disruption of the dimer interface directly results in the monomeric mutants. Nevertheless, the fact that little or no $C_{55}$-P/lipid I copurifies with the monomeric mutants indicates a substantial loss of the interfacial binding sites. Overall, therefore

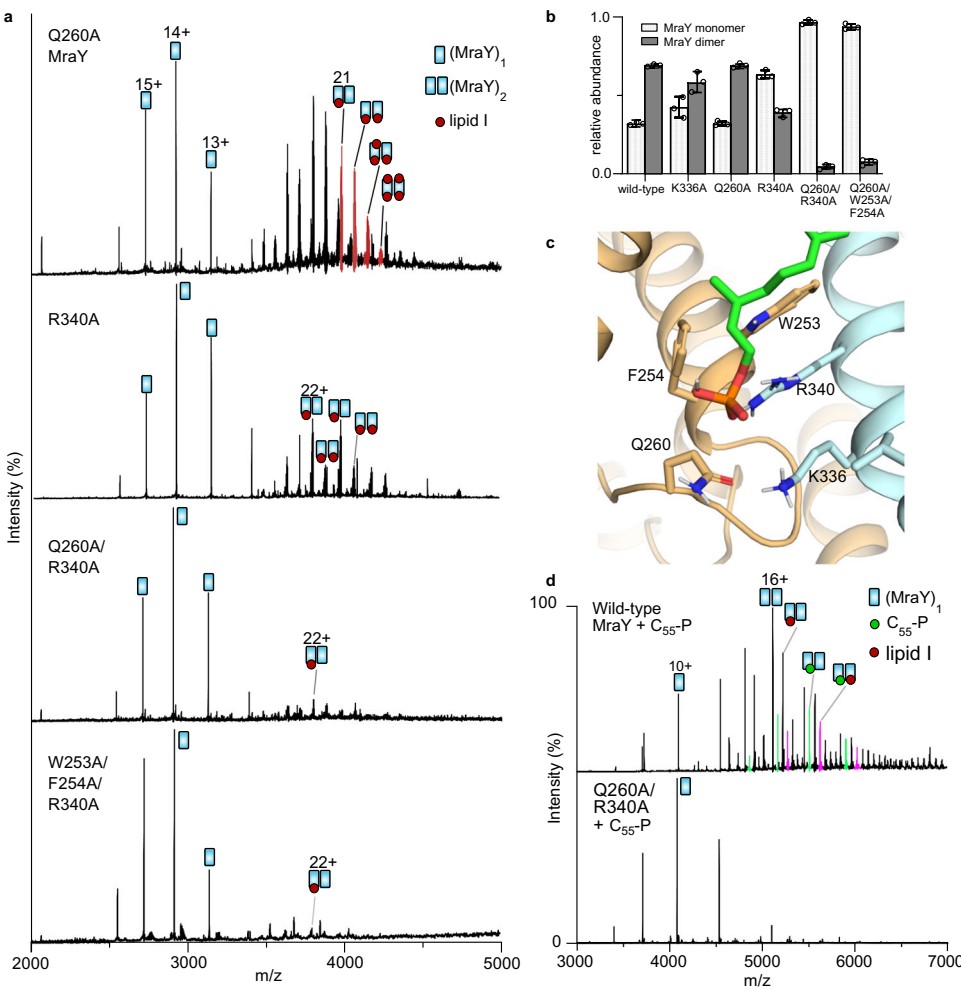

**Fig. 2 MraY binding of carrier lipid is coupled to dimerisation. a** Native mass spectra of MraY mutants. Shown are representative spectra of 5 µM MraY protein variants: Q260A, R340A and Q260A/R340A and W253A/F254A/R340A all liberated from 1% OG micelles. Expression and purification conditions, as well as instrument settings, are identical to those of wild-type MraY (cf. Fig. 1a). Monomer-dimer proportions are consistent for replicate protein preparations. Masses are given in Supplementary Table 2. **b** Relative proportions of monomer and dimer species for the wild type and mutants of MraY. For Q260A, each MraY dimer was detected as a complex with 1–4 molecules of endogenous lipid I and little or no disruption of the dimer was observed relative to the wild type. For R340A, both ligand binding and dimerisation were impaired. For the hybrid mutants (Q260A or W253A/F254A with R340A on the other protomer), predominantly ligand-free monomers were observed in each case. Each protein was expressed and purified at least three times starting with a fresh transformation. The bar represents mean ± SD (*n* = 3), each circle represents individual biological replicates. Source data are provided as a Source data file. **c** Zoom-in view of $C_{55}$-P binding pose (from Fig. 1c) showing $C_{55}$-P and the coordinating residues represented as sticks. **d** Spectra recorded for 5 µM MraY (wild type and Q260A/R340A mutant) upon titration with 50 µM $C_{55}$-P. The $C_{55}$-P molecules bind and increase the relative proportion of dimeric wild-type MraY but little or no binding and/or dimerisation occur in the case of the mutant under the same conditions. Buffer and instrument settings were the same as for the data shown in Fig. 1b. Peaks exclusive to $(MraY)_2(C_{55}$-$P)$ and $(MraY)_2(lipid\ I)(C_{55}$-$P)$ are highlighted, green and pink, respectively.

these results show that disruption of interfacial $C_{55}$-P binding residues is coupled with the destabilization of the MraY dimer.

We next tested whether exogenous $C_{55}$-P can induce dimerisation of wild-type MraY and the Q260A/R340A mutant by incubating the protein variants with a tenfold molar excess of $C_{55}$-P. We recorded mass spectra by releasing the proteins from C8E4 micelles. The resulting spectra show that the wild-type MraY binds to $C_{55}$-P and that the relative populations of dimers were enhanced. In contrast, the Q260A/R340A MraY mutant failed to dimerise and no $C_{55}$-P binding was observed under the same conditions as used for the wild-type protein (Fig. 2d).

To compare directly the $C_{55}$-P binding affinities of wild-type and the mutant MraY, we need first to remove copurified ligands from the wild type and compare both protein variants (wild type and Q260A/R340A) in their monomeric forms. Thus, we buffer-

exchanged the wild-type and the Q260A/R340A MraY into a buffer containing 0.05% LDAO (established as harsher, delipidating detergent than C8E4 or OG)[23] by size-exclusion chromatography and prepared aliquots containing a fixed protein concentration and an increasing $C_{55}$-P concentration. Importantly, the wild-type MraY formed $(MraY)_1(C_{55}$-$P)_1$ and $(MraY)_1(C_{55}$-$P)_2$ complexes with a fourfold molar excess of ligand (Supplementary Fig. 6a). In contrast, only the $(MraY)_1(C_{55}$-$P)_1$ complex was observed for the Q260A/R340A mutant under the same condition (Supplementary Fig. 6b). This difference in binding stoichiometry suggests that only the interfacial $C_{55}$-P binding site was disrupted in the mutant and that $C_{55}$-P can still bind at the proposed catalytic cleft of each MraY protomer. Analysis of the relative abundance of ligand-free and $C_{55}$-P bound proteins shows that the wild type has a

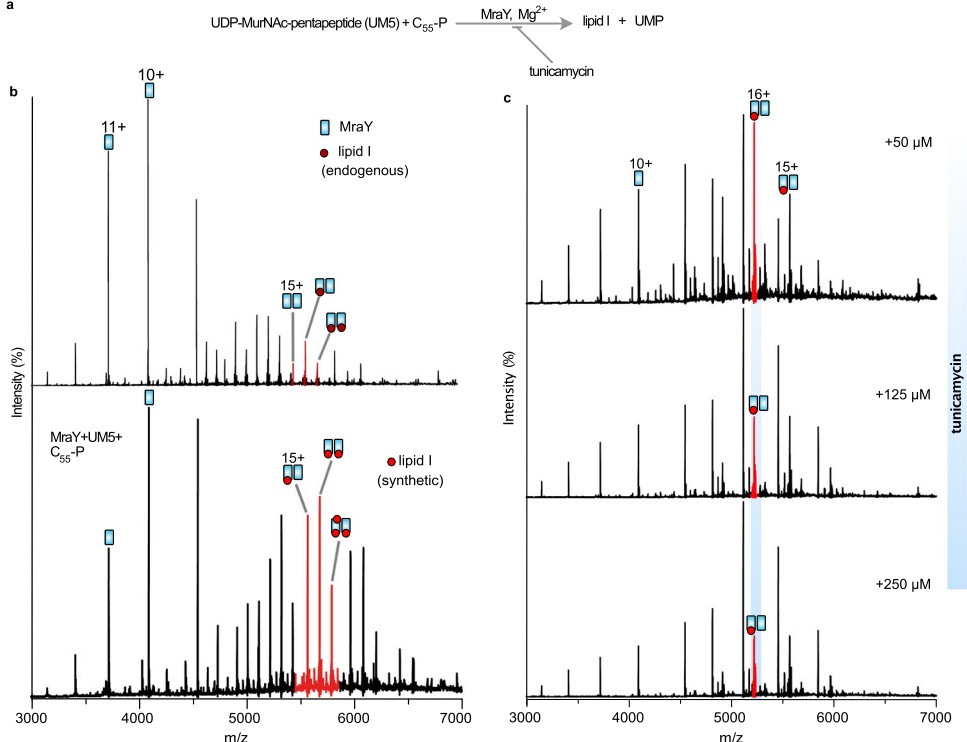

**Fig. 3 Monitoring enzymatic activity and inhibition of MraY. a** A reaction scheme illustrating MraY-mediated synthesis of lipid I and inhibition by tunicamycin. UMP uridine monophosphate. **b** Native mass spectrum of MraY bound to endogenous lipid I (top panel) and incubated with a tenfold molar excess of $C_{55}$-P and UM5 for 16 h (lower panel) in buffer containing 0.02% DDM, 20 mM Tris (pH 8.0), 200 mM NaCl, of 2 mM $Mg^{2+}$ and 10% glycerol. Spectra were acquired after incubation and buffer exchange into 0.5% C8E4, 200 mM ammonium acetate, pH 8.0. Uridine monophosphate (UMP) is released as a by-product (not observed). Although MraY catalysis does not occur at the dimer interface, molecules of lipid I, synthesised in situ, were captured as adducts to the dimer (mass 1674 Da). In the presence of synthetic lipid I, the total intensity of peaks assigned to dimeric MraY is higher than for the protein control without substrates added (cf. Fig. 1b). For clarity, the dimer (15+) and lipid-bound charge states are highlighted (red). **c** The same lipid I synthesis reaction as above but performed in the presence of an increasing concentration of tunicamycin. The number and intensity of synthetic lipid I adducts decrease with increasing tunicamycin concentration. Excess substrate and inhibitor molecules were removed via buffer exchange prior to measurements.

consistently higher affinity for $C_{55}$-P than the Q260A/R340A mutant (Supplementary Fig. 6c). Overall, these data confirm predictions of $C_{55}$-P binding residues from MD simulations and highlight in particular the significant role of R340 in coordinating the $C_{55}$-P molecules bound to MraY at the interfacial sites.

**Probing enzymatic activities of MraY by native mass spectrometry**. The finding that MraY has a strong binding affinity for endogenous lipid I under our experimental conditions prompted us to consider further its enzymatic activity. To this end, we attempted to synthesise lipid I by incubating MraY and its cofactor $Mg^{2+}$ with substrates $C_{55}$-P (848 Da) and UM5 (1149 Da). We choose the Gram-positive form of UM5 with l-Lysine in place of *meso*-2,6-diaminopimelic acid (m-DAP) in the pentapeptide stem such that the resulting lipid I would correspond to 1674 Da and will therefore be distinguishable by mass from the copurified endogenous lipid I (1717 Da). After 16 h of incubation and subsequent buffer exchange into C8E4-containing buffer, we observed a new series of adduct peaks that correspond to lipid I (1674.3 ± 1.1 Da) bound to the MraY dimer (Fig. 3a). Compared to the protein incubated without substrates (UM5 and $C_{55}$-P), the presence of this synthesised exogenous lipid I yielded an increase in the relative proportion of dimers (cf. Fig. 1b). Importantly, lipid I is also formed upon incubation of MraY with UM5 and $Mg^{2+}$ ions but without the addition of exogenous $C_{55}$-P (Supplementary Fig. 7a). These results, therefore, confirm our assignment of endogenous $C_{55}$-P above (Fig. 1a) and demonstrate

that MraY-mediated synthesis of lipid I can be monitored directly by native MS.

Having achieved in situ synthesis of lipid I with detergent purified MraY, we realised that we had a means of probing its inhibition through the application of antibiotics and subsequent study by MS. To this end, we investigated the impact of tunicamycin on lipid I formation. We used a fixed concentration of the starting materials (UM5 and $C_{55}$-P) and an increasing concentration of tunicamycin (50–250 μM) (Fig. 3c). At the lowest concentration of tunicamycin tested, the number and intensity of lipid I adducts decreased compared with spectra recorded in the absence of the antibiotic. Increasing tunicamycin concentrations to 125 and 250 μM led to further depletion of the lipid I product reflecting effective inhibition of MraY-mediated lipid I synthesis by tunicamycin. Thus, we have captured enzymatic activities of MraY and this approach can be used to monitor antibiotic inhibition of enzymes that exhibit a measurable binding affinity for their synthetic product.

**MurG binds lipid I with a high affinity**. Next in function to MraY in the PG biosynthetic pathway is MurG, a peripheral protein on the inner leaflet of the cytosolic membrane that transfers a GlcNAc molecule to lipid I to form lipid II. We purified *E. coli* MurG to investigate its substrate-binding properties and enzymatic activity by native MS. When released from a buffer containing 0.05% LDAO, 200 mM ammonium acetate (pH.8.0), the resulting spectrum of MurG displayed charge state

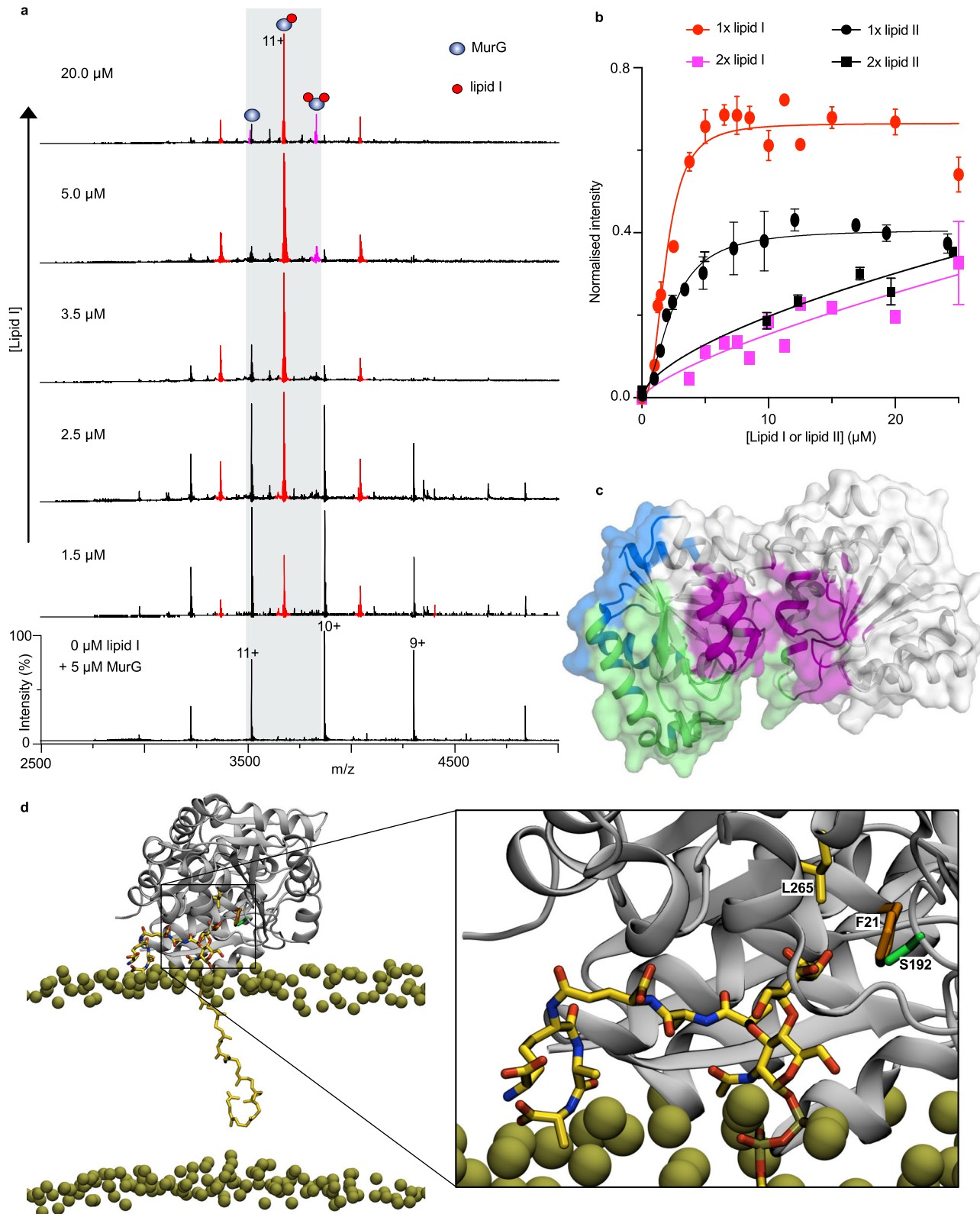

series corresponding to monomers (Fig. 4a). To investigate the affinity of MurG for its native substrate lipid I, we recorded spectra for solutions containing 5 μM MurG and 0–25 μM lipid I (Fig. 4a). At lipid I concentration of 1.5 μM, charge state series were observed with an additional mass of 1717 Da, consistent with one molecule of lipid I bound to MurG. Further increase in lipid I concentration yielded additional charge state

reflecting binding of a second lipid I molecule to MurG. Notably at MurG: lipid I molar ratios ≥1, the MurG-lipid I complex becomes predominant over the unbound form of MurG. A fit to the relative intensity of lipid I-bound MurG at different concentrations for the first lipid I binding event yielded an apparent dissociation constant ($K_d$) of 1.9 ± 0.4 μM and a Hill coefficient of h = 2.6 ± 1.2 (Fig. 4b). This indicated that lipid I molecules bind

**Fig. 4 MurG-lipid interactions. a** Mass spectra of 5 μM MurG in the presence of increasing concentrations of lipid I. At 1.5 μM lipid I, charge state series corresponding to protein-bound lipid I molecules are observed and their population increases with increasing lipid I concertation. **b** Intensities of lipid I- or lipid II-bound MurG over the total peak intensities as a function of lipid concentrations. For MurG-lipid I binding, a fit to the data corresponding to the first binding event (red circles) yielded $K_d = 1.89$ μM (95% CI: 1.53–2.35), $B_{max} = 0.66$ (95% CI: 0.62–0.72), h = 2.6 (95% CI: 1.7–4.0) and $r^2 = 0.94$. For MurG-lipid II binding (black circles), $K_d = 2.64$ μM (95% CI: 2.04–3.79) and $B_{max} = 0.39$ (95% CI: 0.35–0.46), h = 1.8 (95% CI: 1.1–2.8), $r^2 = 0.97$. Each data point is the mean (± SD) (n = 3) of three independent measurements. Error bars might not be visible when smaller than the symbol. Source data are provided as a Source data file. **c** Primary lipid I and lipid II binding sites as identified from 10 × 15 μs CG data. All sites with >50% occupancy are shown. Two sites on the amphipathic helix are shown in blue ($k_{off} = 0.46 \pm 0.15$ μs$^{-1}$ for lipid I and $0.40 \pm 0.34$ μs$^{-1}$ for lipid II, based on 50 rounds of bootstrapping) and green ($k_{off} = 0.14 \pm 0.25$ μs$^{-1}$ for lipid I and $0.94 \pm 0.53$ μs$^{-1}$ for lipid II). The site near the ligand-binding region (purple) has $k_{off} = 0.70 \pm 1.03$ μs$^{-1}$ for lipid I and $1.91 \pm 0.30$ μs$^{-1}$ for lipid II. Little or no binding is depicted as white (<50% occupancy). **d** View of lipid II bound to MurG at the substrate-binding site as identified using CG and simulated with atomistic MD. Relevant RMSDs are shown in Supplementary Fig. 9d. Lipid II is shown as yellow, red and blue sticks and the membrane phosphates as gold spheres. The protein is shown as a grey cartoon, with the GlcNAc-coordinating residues as coloured sticks: Phe-21 in orange, Ser-192 in green and Leu-265 in yellow.

to MurG with a high affinity and at more than one site with positive cooperativity. Equivalent analysis of MurG-lipid II binding interactions yielded a higher apparent $K_d$ of $2.6 \pm 0.9$ μM (Fig. 4b and Supplementary Fig. 8) than for lipid I. Overall, these data indicate that MurG binds its native substrate lipid I with a higher affinity than its glycosylated product lipid II.

Further insights into the interactions of MurG with lipid I and lipid II molecules were gained through CG MD simulations. We initiated simulations with monomeric MurG positioned *ca.* 10 nm from a model membrane containing either lipid I or lipid II. We then ran the simulations for 15 μs to allow MurG to bind to the membrane. Analysis of the binding data with PyLipID reveals three prominent binding sites for lipid I and lipid II on MurG: two of these sites are in close proximity, centred on the helix from Lys-72 to Lys-93 (Fig. 4c), and were previously predicted to be important for MurG interaction with the membrane[14]. A third site also exists, nearer to the structurally-resolved substrate-binding site[29] (Fig. 4c; magenta). This site involves the peptide and sugar headgroups of lipid I and lipid II making extensive contact with the entire substrate-binding region (Supplementary Fig. 9). Interestingly, one of the primary lipid II binding poses involves the GlcNAc of lipid II interacting with several of the GlcNAc-coordinating residues in the cocrystal structure of MurG;[29] these residues include Phe-21, Ser-192 and Leu-265 (Fig. 4d). We then converted a bound CG pose to an atomistic one and performed a set of MD simulations, which further supported the stability of this binding orientation (Supplementary Fig. 9). This pose might represent an enzymatic intermediate state, whereby GlcNAc has just been added to lipid I to form lipid II, which will now be released from the active site. Across the CG MD data, we observed on average $2.15 \pm 0.5$ lipid I molecules bound with high affinity to MurG sites, in reasonable accord with the MS data. Analysis of $k_{off}$ from these simulations reveals that lipid II is less strongly bound to MurG than lipid I by a factor of about 2-3 ($k_{off} = 0.70 \pm 1.03$ μs$^{-1}$ for lipid I and $k_{off} = 1.91 \pm 0.30$ μs$^{-1}$ for lipid II) at this site. The MD results are consistent with the apparent $K_d$ determined by native MS experiments above.

**Passage of peptidoglycan precursors is driven by enzyme-substrate affinity gradients.** Since lipid I is a natural substrate of MurG we reasoned that MurG must exhibit a higher affinity for lipid I than MraY. To test this hypothesis, we included MurG in our lipid I synthesis reaction (MurG, MraY, Mg$^{2+}$, C$_{55}$-P and UM5) to allow competition of both proteins for lipid I molecules during the in situ synthesis reaction. Accordingly, after incubation, the mass spectra reveal charge states corresponding to apo MraY, apo MurG, as well as peaks corresponding to lipid I binding to both proteins (Fig. 5b). Whereas MraY was more abundant than MurG in solution (3:1 molar ratio), we found that

only ~30% of MraY formed a complex with lipid I with a 1:1 stoichiometry. The majority of MraY ~70% remained ligand-free. In contrast, ~85% of MurG had formed a complex with lipid I, leaving only 15% in a ligand-free status. This relatively high-affinity binding to MurG, as opposed to MraY, is consistent with lipid I being a natural substrate of MurG. Furthermore, MurG formed both 1:1 and 1:2 stoichiometric complexes with lipid I or lipid II, suggesting the possibility of two substrate-binding sites per MurG monomer.

Having demonstrated preferential binding of lipid I to MurG over MraY, the next step in the pathway is to form lipid II from lipid I. We decided to do this in a concerted way: first to form lipid I as above (with C$_{55}$-P, Mg$^{2+}$ and UM5) in the presence of MraY and MurG and then to add UDP-GlcNAc to form lipid II. We incubated the two membrane enzymes MurG and MraY in their DDM micelle environments with the components (C$_{55}$-P, UM5, and UDP-GlcNAc) and recorded the mass spectra after buffer exchange into LDAO such that both enzymes could be studied under the same solution conditions (See Methods, Fig. 5c). Peaks in this spectra can be assigned to apo monomeric MraY and MraY:lipid II as well as to apo MurG, MurG:(lipid I)$_{1\&2}$, MurG:(lipid II)$_{1\&2}$ and MurG:(lipid I, lipid II). The ternary complex MurG:(lipid I, lipid II) could either be an intermediate in the synthesis or a complex formed after completion of the reaction whereby a new substrate lipid I has been acquired before the full release of a lipid II product. This result highlights the capacity of MurG to simultaneously bind both lipids with a similar affinity. To further confirm the identity of the endogenous ligand bound to MraY, we incubated MraY with MurG and UDP-GlcNAc and observed the formation of lipid II (Supplementary Fig. 7b–d). This further confirms our initial assignment of the endogenous substrate bound to MraY as lipid I.

The flippase MurJ is responsible for translocating lipid II across the inner membrane and has been previously shown to bind lipid II with high affinity[16]. We, therefore, hypothesised that MurJ would exhibit a markedly higher affinity for lipid II than MraY or MurG. To this end, we prepared an aliquot of the lipid II synthesis reaction as described above (MurG, MraY, C$_{55}$-P, UM5 and UDP-GlcNAc) and then added MurJ such that the three membrane enzymes are in the molar ratio of 3:1:3 (MraY:MurG:MurJ). This ratio was found to be optimal to observe all three proteins simultaneously with similar intensities in the mass spectrum using the same instrument settings. Intriguingly, the spectrum showed that MurJ formed an intense 1:1 stoichiometric complex with lipid II, leaving MraY and MurG in their apostates (Fig. 5d). These observations imply that MurJ has a higher affinity for lipid II, its natural substrate, than either MurG or MraY.

To validate these results, we assessed the relative affinities of MraY, MurG and MurJ for their respective lipid substrates individually and compared the products in a pair-wise manner.

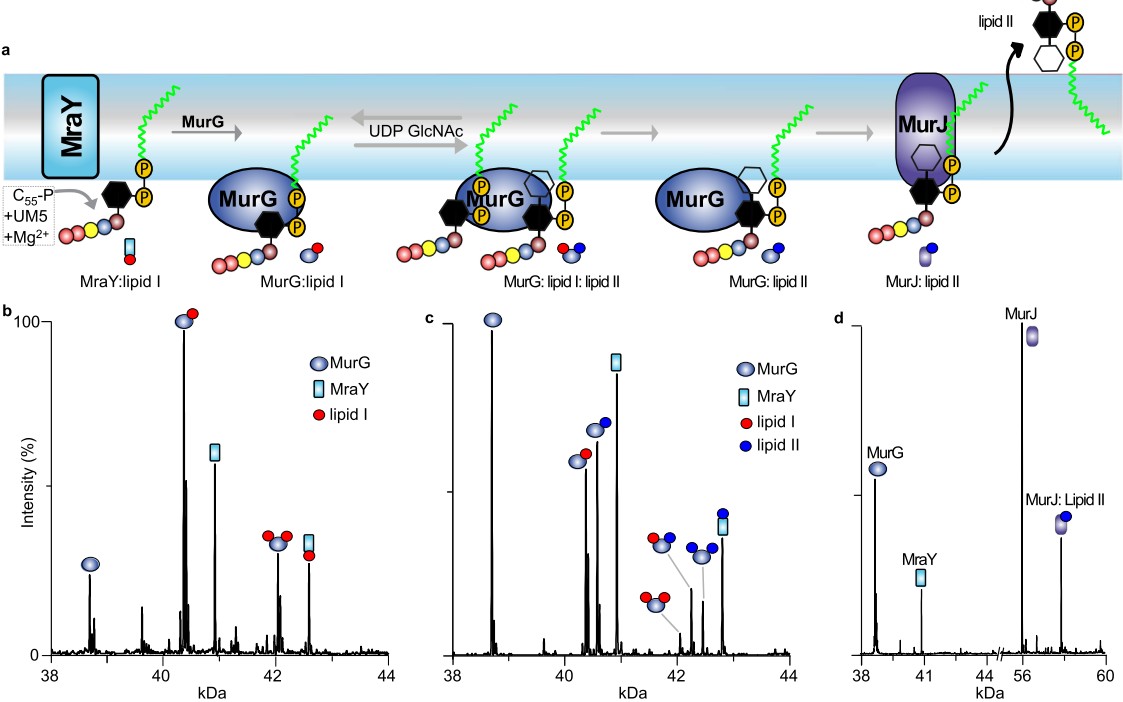

**Fig. 5 Coupled synthesis of lipid II by MraY and MurG and its passage to MurJ. a** Schematic illustrations showing the steps involved in the coordinated synthesis of lipid II and its passage to the flippase MurJ. For simplicity, MraY is shown here as monomeric. **b** Deconvoluted mass spectrum of MurG included in a lipid I synthesis reaction (MraY incubated with $Mg^{2+}$, $C_{55}$-P and UM5). MurG forms 1:1 and 1:2 stoichiometric complexes with synthesised lipid I and binds lipid I with higher affinity than MraY, even though MraY is in threefold molar excess. **c** Deconvoluted mass spectrum capturing the coupled synthesis of lipid I and lipid II following incubation of MurG and UDP-GlcNAc with MraY, $Mg^{2+}$, $C_{55}$-P and UM5. Both lipid I and lipid II (synthesised during the reaction) bind to MurG but only lipid II forms adducts with MraY under these conditions. **d** Competition amongst MurG, MraY and MurJ for lipid II, the latter being synthesised in situ following incubation of MurG and MraY with UDP-GlcNAc, $Mg^{2+}$, $C_{55}$-P and UM5 prior to addition of MurJ. The fact that MurJ binds all available lipid II indicates that the MurJ flippase has a higher affinity for lipid II than either MurG or MraY. Spectra were obtained under the same conditions by releasing the protein from LDAO micelles where the three proteins are stable. All reactions were however carried out in a DDM-containing buffer where MraY is an equilibrium of monomer and dimer. Observed masses are listed in Supplementary Table 2.

To enable direct comparison, the membrane enzymes were prepared in the same buffer (0.5% LDAO, 200 mM ammonium acetates, pH 8.0) and spectra were recorded using the same instrument settings (activation of 100 V). The resulting data for MraY and MurG show that $C_{55}$-P binds more favourably to MraY (Supplementary Fig. 10a, b). This is in accordance with the fact that $C_{55}$-P is the native substrate of MraY. Similarly, we find that lipid I binds to MurG with higher intensity than to MraY (Supplementary Fig. 10c, d), indicating that a MraY protomer readily releases lipid I for further processing by MurG. Analogously lipid II binds to the flippase MurJ with a higher affinity than to MurG, the preceding enzyme in the pathway (Supplementary Fig. 10e, f). These observations are consistent with the binding gradients observed in our in situ competitive assay reaction (Fig. 5). Together these results validate our hypothesis that differential binding affinities of carrier lipid for these enzymes present a driving force for their onward passage to the flippase MurJ for export across the bacterial cytoplasmic membrane to enable the maturation of a nascent PG.

## Discussion

We have employed native MS to study the interactions of lipid substrates with MraY, MurG and MurJ (summarised in Fig. 6). Briefly, our MS data show that MraY can exist in a monomer-dimer equilibrium and that dimers bind $C_{55}$-P and/or lipid I more efficiently than monomers with protein: lipid-binding stoichiometry of ~2:4. Our MD simulations and mutagenesis experiments indicate that $C_{55}$-P can interact with MraY at the dimer interface. The formation of 1:1 and 1:2 stoichiometric MurG-lipid I/lipid II complexes suggests the possibility of two lipid-binding sites per MurG monomer, implying a possible role for this enzyme in coordinating substrate and product release. MurJ captured all lipid II produced in the MraY/MurG coupled reactions, indicating that MurJ has the highest affinity for lipid II of these three membrane enzymes.

Considering our findings in detail, our data provide a strong indication that the oval-shaped hydrophobic tunnel at the dimer interface of MraY[17] is large enough to accommodate at least one molecule of $C_{55}$-P for each subunit at any given time. Previous studies indicate that the negatively charged exogeneous dimyristoyl phosphatidylglycerol lipid enhances dimer formation in MraY[24,30], suggesting that electrostatic interactions of the ligand headgroup play a key role in stabilising the dimer interface. However, the relatively shorter hydrophobic tail of phospholipids, in comparison to $C_{55}$-P and lipid I, imply that phospholipid molecules are less likely to provide the same depth of hydrophobic contacts as the undecaprenyl tails that could span the dimer interface. It has been speculated that the dimer interface of MraY is filled with phospholipids or the carrier lipid molecules[17,24]. Whereas there is growing evidence that interfacial lipids are crucial for stabilising membrane protein oligomers[31,32], stabilisation of the MraY dimer by $C_{55}$-P and lipid I is intriguing in that it is an oligomer mediated by its own lipid substrate and catalysed product.

The finding that MraY binds to endogenous $C_{55}$-P and lipid I with an affinity such that the bound ligands survive detergent

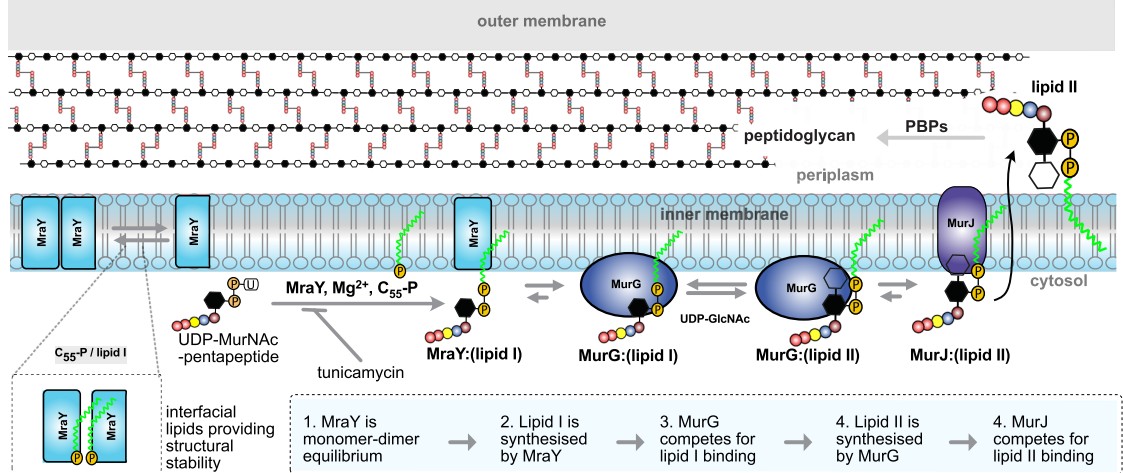

**Fig. 6 Insights into key steps of peptidoglycan biosynthesis from native MS.** MraY exists in a monomer-dimer equilibrium with dimerisation favoured in the presence of endogenous $C_{55}$-P and lipid I. MraY binds lipid substrates to stabilise its dimeric state. Synthesis of lipid I via MraY can be effected in vitro from $C_{55}$-P and UDP-MurNAc-pentapeptide, and a change in mass confirms that endogenous lipid I was displaced by its biosynthetic counterpart. Tunicamycin binds to MraY and inhibits lipid I formation. MurG binds to lipid I with a higher affinity than MraY, to drive the passage of lipid I along the pathway, and forms lipid II in the presence of UDP-GlcNAc. The flippase MurJ then outcompetes MurG and MraY to bind lipid II, forming the binary 1:1 complex. Lipid II is flipped into the periplasm where its polar disaccharide peptide head is incorporated into the peptidoglycan polymer.

solubilisation and purification is surprising, considering the limited number of lipid carrier molecules ($1.5 \times 10^5$) in the membrane[33] and their rapid passage ($\sim$90 s)[34] in the closed-loop of the lipid II cycle. The catalytic aspartate residues on each MraY protomer[17,18,35] are remote from the dimer interface and distinct from the $C_{55}$-P coordinating residue (R340) reported in this study. Taking into account the molecular structure of the MraY-tunicamycin complex[18], $C_{55}$-P was shown to interact with MraY close to the groove surrounding TM9b. How $C_{55}$-P is recruited from the membrane to the proposed site remains unclear in the absence of a MraY structure with $C_{55}$-P or a similar precursor. The hydrophobic chain of $C_{55}$-P extends beyond the bilayer thickness of the membrane and suffers from limited rotational freedom, imposed by the isoprene units. Therefore, $C_{55}$-P bound to MraY with its tail at the shallow groove, proposed to accommodate the tunicamycin tail, likely bends to allow the phosphate headgroup to access the coordinating aspartate residue. This is consistent with our MD simulation data that reveal relatively weaker $C_{55}$-P binding near the proposed active site than at the interfacial sites. While this indicates that MraY can interact with $C_{55}$-P at multiple sites with different affinities, it raises a question regarding the role of $C_{55}$-P molecules whose phosphate groups bind adjacent to the dimer interface. Given the observed connection between dimerisation and the interfacial binding of endogenous $C_{55}$-P and lipid I, binding of these molecules at the interfacial sites likely provides structural stability to MraY dimer. We therefore ascribe structural roles to the interfacial lipids bound to MraY dimers.

Our MS data suggest that MurG functions as a monomer, at least in vitro. The formation of 1:1 and 1:2 stoichiometric MurG:lipid I complexes also suggests the possibility of two substrate-binding sites per MurG monomer. Consistent with these results, we identified several potential binding sites for lipid I on MurG using MD simulations, including one in which the MurNAc moiety is proximal to the location of GlcNAc in the x-ray crystal structure[29]. This is particularly clear when we run simulations of lipid II binding to MurG, wherein we observed interactions with the same residues of MurG that coordinate with GlcNAc in the x-ray crystal structure. We propose that this site could represent the binding region of lipid I for optimal synthesis of lipid II. Additionally, we showed by native MS that MurG can bind

simultaneously to lipid I and lipid II, implying a possible coordinating role in the synthesis and release of the lipid II product. We also found that MurG binds to lipid I in preference to the monomeric form of MraY, and that MurJ exhibited a higher affinity for lipid II than MurG, the preceding enzyme in the pathway. This is an indication that the relative affinity of lipid substrates for their respective downstream enzymes presents a driving force for the lipid II precursor towards the flippase MurJ for onward translocation across the inner membrane.

In summary, rather than the traditional radiochemical labelling or extraction of lipid analytes for analysis by a thin layer or high-performance liquid chromatography[13,17,21,36], we take advantage of the noncovalent enzyme-ligand associations as a qualitative, in situ means of capturing details of precursor biosynthesis and interactions. By studying three membrane enzymes with multiple substrates, we have demonstrated the potential for following biosynthetic reactions using native MS and revealed the relative binding affinities that likely drive precursor passage during peptidoglycan synthesis. From a methodological standpoint, results from this study highlight how native MS, in conjunction with MD, can complement other biophysical approaches. By defining relative binding affinities for different lipid substrates and products, native MS can inform high-resolution structures wherein density for flexible hydrophobic ligands can be difficult to define. Moreover, being able to follow key biosynthetic reaction sequences in such detail offers new opportunities for antibiotic discovery and for investigating their mechanisms of action. For example, the development of new compounds that perturb the monomer-dimer equilibrium of MraY, or that selectively target the binding sites of MurG, could pave the way for new anti-microbials. Overall, due to the variety of roles of the carrier lipid in cellular metabolism[37], the development of lipophilic molecules that tilt binding affinity gradients of PG precursor lipids away from their respective biosynthetic enzyme presents new opportunities for cell wall directed interventions to combat anti-microbial resistance.

## Methods
**Chemicals**. *n*-decyl-$\beta$-D-maltoside (DM), *n*-dodecyl-$\beta$-D-maltoside (DDM), *n*-nonyl-$\beta$-D-glucopyranoside (NG), *n*-octyl-$\beta$-D-glucopyranoside (OG), 2,2-dihexylpropane-1,3-bis-$\beta$-D-glucopyranoside (OGNG), and *n*-dodecyl-*N*,*N*-dimethylamine-*N*-oxide

(LDAO) and tetraethylene glycol monooctyl ether (C8E4) were from Anatrace (Maumee, USA). UDP-MurNAc-pentapeptide (version with L-Lys) and lipid I were purchased from the UK-BACWAN facility (University of Warwick, UK). Undecaprenyl phosphate ($C_{55}$-P) was from Larodan (Monroe, USA). The m-DAP variant of lipid II[38,39] was a generous gift from Eefjan Breukink (Bijvoet Centre for Biomolecular Research, University of Utrecht).

**Plasmids**. The plasmid pET26-MBP-MraYaa bearing HRV3C protease cleavage site between the MBP and the wild-type MraY was a kind gift from S-Y Lee (Addgene plasmid #100166)[17]. Plasmids of MraY mutants were generated using oligonucleotide-directed mutagenesis using pairs of primers listed in Supplementary Table 5. Plasmids for expressing *E. coli* MurG and *E. coli* MurJ were generated on a modified pET15b vector and contain a C-terminal GFP-His₆ and His₆ fusions, respectively.

**Protein expression and purification**. For MraY (wild type and mutants), the plasmid was transformed into chemically competent *E. coli* C41(DE3) cells (Lucigen). Plasmids for MurG and MurJ were transformed into C43(DE3) *E. coli* cells. Cells were grown at 37 °C to OD$_{600nm}$ 0.6–0.8, and protein expression was induced by the addition of 1 mM IPTG. After 4 h at 37 °C, cells were harvested by centrifugation at 5000 × *g* and stored at −80 °C until required. Cell paste was thawed on ice and resuspended in a buffer containing 20 mM Tris-HCl, pH 8.0, 150 mM NaCl, and EDTA-free protease inhibitor cocktail tablets (Roche). The cells were then disrupted by 4–5 passes over a microfluidizer (Microfluidics) at 20,000 psi. Non-lysed cells and debris were pelleted by centrifugation at 20,000 × *g* for 20 min and the clarified lysate was retained.

MBP-MraY was extracted from the clarified lysates by incubation with 1% DDM for 60 min. Nonsolubilised aggregates were removed by centrifugation at 20,000×*g* for 1 h and the supernatant was loaded on a 5-mL HisTrap HP column preequilibrated in buffer A (20 mM Tris-HCl (pH 8.0), 200 mM NaCl, 0.03% DDM, and 40 mM imidazole). After washing with buffer B (20 mM Tris-HCl (pH 8.0), 200 mM NaCl, 0.03% DDM, and 80 mM imidazole), the protein was eluted from the column with a linear gradient of buffer C (20 mM Tris-HCl, pH 8.0, 200 mM NaCl, 0.03% DDM, and 500 mM imidazole). MBP-MraY was pooled, passed over a PD-10 desalting column (GE Healthcare), and digested overnight with HRV3C protease (Merck). MraY was isolated from the digestion mixture by size-exclusion chromatography (SEC) using a Superdex S200 increase column. The SEC buffer was 20 mM Tris-HCl (pH 8.0), 200 mM NaCl, 0.025% DDM. The protein was concentrated to 65 µM, aliquoted, flash-frozen in liquid nitrogen, and stored at −80 °C. All MraY variants (wild type and mutants) were expressed and purified at least three times under the same conditions, starting with single colonies from a freshly transformed C41(DE3) host strain.

MurG-GFP was extracted from cell lysates in a buffer containing 0.5% DDM and purified over a 5-mL HisTrap HP column. MurG-GFP was treated with tobacco etch virus (TEV) protease and dialysed overnight. The mixture was subsequently incubated with nickel nitriloacetic acid (NTA) agarose resin and MurG was collected as the flow-through. The protein was finally cleaned up by SEC as previously described. MurJ were purified as previously described[16]. Briefly, the clarified cell lysates were ultra-centrifuged at 140,000×*g* to pellet the membranes. The proteins were solubilized from the membrane fractions with 20 mM Tris-HCl, pH 8.0, 150 mM NaCl, 20% glycerol supplemented with 2% DDM for 2 h for UppP and with 2% DDM and 2% OGNG for 16 h for MurJ at 4 °C. The insoluble matrix was removed by centrifugation and the supernatant was purified over 5-mL HisTrap HP column. Proteins were concentrated, passed over the PD-10 column, and finally purified by SEC. Proteins were either used immediately or flash-frozen in liquid nitrogen and stored at −80 °C. Protein concentration was measured using a Biomate UV detector at 280 nm using extinction coefficients calculated from their predicted amino acid sequences.

**Sample preparation for native MS**. $C_{55}$-P was dissolved in methanol at a final concentration of 1 mM. Prior to use, methanol was removed under a gentle stream of nitrogen and the lipid film resuspended in SEC buffer. Stock solutions of 1 mM tunicamycin and 1 mM UM5, were made in SEC buffer. For the ligand-binding experiments, the protein was incubated with the desired concentration of ligand. After overnight incubation at 4 °C, the mixture was buffer-exchanged into MS buffer and diluted then to a final protein concentration of 10 µM before measurements. Lipid I reaction mixture was made by incubating 30 µM MraY, 100 µM $C_{55}$-P, 100 µM UDP-MurNAc-pentapeptide, and 5 mM MgCl₂ in a buffer containing 20 mM Tris-HCl (pH 8.0), 200 mM NaCl, 0.02% DDM. Lipid II reaction mixture was made as for lipid I but the mixture, in addition, contained 30 µM MurG and 100 µM UDP-GlcNAc. Samples were incubated overnight at 4 °C and then exchanged into 0.05% LDAO, 200 mM ammonium acetate ("MS buffer") using a centrifugal buffer exchange device (Micro Bio-Spin 6, Bio-Rad). For the in situ concerted reactions, the proteins were buffer-exchanged into LDAO as it provides sufficient stability for all the three enzymes MraY, MurG and MurJ. Of note detergents used to study MraY (C8E4 and OG) caused precipitation of MurG and MurJ.

**Native mass spectrometry**. Mass spectra were acquired on a Q-Exactive hybrid quadrupole-Orbitrap mass spectrometer (Thermo Fisher Scientific, Bremen, Germany) optimised for transmission and detection of high molecular weight protein complexes[40]. About 3 µl of protein aliquot was transferred into gold-coated borosilicate capillary (Harvard Apparatus) prepared in-house and capillary and mounted on the nano ESI source. The instrument settings were 1.2 kV capillary voltage, S-lens RF 200%, quadrupole selection from 1000 to 20,000 m/z range, argon UHV pressure 3.3 × 10⁻¹⁰ mbar, capillary temperature 200 °C, resolution of the instrument was set to17,500 at a transient time of 64 ms. Voltages of the ion transfer optics –injection flatapole, inter-flatapole lens, bent flatapole, and transfer multipole were set to 5, 3, 2, and 30 V respectively. The noise level was set at 3. Unless otherwise stated all the experiments were performed in the positive polarity. For the CID MS/MS experiments to detect the ligands bound to proteins, negative polarity was used and the instrument ion optics were tuned accordingly to enhance transmission and detection of low m/z ions. Theoretical isotopic distributions of $C_{55}$-P and lipid I were calculated using an online tool (https://prospector.ucsf.edu/prospector/cgibin/msform.cgi?form=msisotope). Data were visualised and exported for processing using the Qual browser of Xcalibur 4.1.31.9 (Thermo Scientific) and spectral deconvolution was performed using UniDec software[41]. Relative binding affinities were obtained from deconvoluted spectra by dividing the intensity of ligand-bound protein peaks by the sum of the intensities of ligand-bound and ligand-free proteins peaks. All measurements were performed at least three times and yielded similar results.

**Lipid I and lipid II binding to MurG**. Lipid I and Lipid II binding experiments were performed with the protein in 200 mM ammonium acetate supplemented with 0.05% (w/v) LDAO. To obtain the binding constant for the interaction between lipid I and MurG, lipid I was added in increasing amounts while keeping the protein concentration the same at 5 µM. The same approach was adopted for the binding of lipid II to MurG. Spectra were deconvoluted using UNIDEC to extract peak intensities. The ratios of the intensity of the ligand-bound species with respect to the total intensity of bound and unbound proteins were calculated for binding of the first and second ligand molecule. The mean and standard deviation of these fractional binding intensities from three independent experiments were plotted against lipid I concentration. The data were fitted globally using GraphPad Prism 9.0 to the Eq. (1):

$$y = c\frac{B_{max}x^h}{(x^h + K_d{}^h)} \tag{1}$$

where $B_{max}$ is the maximum specific binding, $K_d$ is the apparent dissociation constant, and h is the Hill coefficient.

For the binding of second molecule, we could not deduce the low-affinity $K_d$ because a much higher concentrations of lipid I or lipid II caused a drastic reduction in spectral quality and potential interference from non-specific binding events through the electrospray process.

**Relative binding affinity of $C_{55}$-P, lipid I, and lipid II towards MraY, MurG, and MurJ**. The proteins were buffer-exchanged into 200 mM ammonium acetate supplemented with 0.05% (w/v) LDAO. Concentrations were determined and 10 µM aliquots were prepared by dilution with the same buffer. Stock solutions of 50 µM $C_{55}$-P, 10 µM lipid I, and 10 µM lipid II were also prepared in the same buffer. Samples were prepared by mixing volumes of desired protein and ligand such that the final protein concentration in each case is 5 µM, $C_{55}$-P at 20 µM, lipid I at 5 µM, lipid II at 5 µM. The same optimised instrument settings (see below) with 100 V in the HCD cell were used to measure all the samples. All experiments were repeated three times from newly prepared stock solutions. Standard deviations were calculated from at least five observed charge states in three independent experiments.

**CG MD simulations**. Coarse-grained simulations were built using PDBs 5CKR in a dimer state (MraY) or chain B of 1NLM (MurG). All non-protein atoms were removed. The proteins were converted to Martini 2.2 using the martinize method, and 1000 kJ mol⁻¹ nm⁻² elastic networks were applied between backbone beads within 1 nm. This helped stabilise the secondary and tertiary structures of each system, and in the case of MraY, stabilise the dimer and stop the dimer interface from collapsing. For MraY, the protein was then built into a mixed symmetric membrane with 65% palmitoyl-2-oleoyl-*sn*-glycero-3-phosphoethanolamine (POPE), 24% palmitoyl-2-oleoyl-*sn*-glycero-3-phosphoglycerol (POPG), 10% cardiolipin using the insane protocol[42], with ca. 1% of either $C_{55}$-P (in a -1 charge state) or lipid I. For MurG, the protein was randomly oriented in 3D space and then a membrane was built 5 nm from the protein. The lipid composition of the model membrane was as for MraY, but with either 1% lipid I or 1% lipid II. The concentration of substrate/product was chosen to match the biological concentration of lipid II of about 1%[43]. For MraY, additional systems were built with 62% POPE, 23% POPG, 10% cardiolipin, and 2.5% of both $C_{55}$-P and lipid I.

$C_{55}$-P parameters were as per previous reports[44] (where it is termed UDP), and parameters for lipid I and II were produced for this study, see Supplementary Fig. 9e for mapping details. The tails of these were based on $C_{55}$-PP from ref. [44], for which the bonded terms were based on atomistic MD of $C_{55}$-PP in a POPE membrane.

The lipid I and II headgroup parameters were taken from peptidoglycan[45]. To finalise the molecules, additional angle terms were applied between the headgroup and tails, as per Supplementary Fig. 9e. These were based upon atomistic MD simulation (see below). Virtual CG beads were imposed on the atomistic system by clustering groups of atoms according to our mapping scheme. Angles between "beads" were then computed using gmx gangle. These values were used to define the lipid II CG parameters, with force constants set to ensure the CG and AT distributions matched (Supplementary Fig. 9e).

Built systems were solubilised with Martini waters and ions to a neutral charge. Systems were minimised using the steepest descent method, then equilibrated using 5 fs time steps for 1 ns, then 20 fs time steps for 10 ns, using a semi-isotropic Berendsen barostat[46] at 1 bar, and a velocity-rescaling thermostat[46] at 323 K. Production simulations were run using the Parrinello–Rahman barostat 1 bar[47], with 20 fs time steps. For each system, 15 μs simulations were run, with $n = 5$ for MraY and $n = 10$ for MurG. For MraY simulations with both $C_{55}$-P and lipid I, five simulations of 60 μs were run. All simulations were run using Gromacs 2020[48].

Identification of lipid-binding sites was performed following a kinetic analysis of the protein–lipid interactions[49] with a programme freely available at https://github.com/wlsong/PyLipID[28]. A double cut-off of 0.55 and 0.8 nm was used. For $C_{55}$-P analyses, only the phosphate and first isoprenyl units were used for the analysis. For the lipid I and II data, all of the headgroup sugar and amino acids were analysed, apart from in the system with both $C_{55}$-P and lipid, where only the phosphate and first isoprenyl unit was used (Supplementary Fig. 4d). Quantification of the number of $C_{55}$-PP bound to lipid I and II was done using the Gromacs tool gmx select.

**Atomistic MD**. Selected poses from the CG data which matched the optimal binding modes calculated by PyLipID were converted to an atomistic description using CG2AT2: https://github.com/owenvickery/cg2at and https://zenodo.org/record/3890164 (ref. [50]). Poses were selected for MraY bound to $C_{55}$-P and/or lipid I, and for MurG bound to lipid II. These poses were converted to the CHARMM36 force field[51]. Alternatively, systems were built containing one lipid II molecule in a POPE:POPG (80:20) membrane, using CHARMM-GUI[52,53]. Each system was minimised using the steepest descents method, then equilibrated with positional restraints on heavy atoms for 100 ps in the NPT ensemble at 310 K with the V-rescale thermostat and semi-isotropic Parrinello–Rahman pressure coupling[47]. Production simulations were run using 2 fs time steps for 3 × 300 ns (MraY-$C_{55}$P and MraY alone ($C_{55}$-P removed)), 160 ns (MurG-lipid II), 3 × 100 ns (lipid II in membrane) or relaxation for 1 ns (MraY-lipid I).

**Reporting summary**. Further information on research design is available in the Nature Research Reporting Summary linked to this article.

## Data availability

Mass spectrometry data, post-equilibration frames used to seed the MD simulations, coordinates and topology files for the lipid substrates and the simulation parameter generated in this study have been deposited to Figshare at https://doi.org/10.6084/m9.figshare.19403852.v2[54]. Systems are coarse-grained unless marked with an 'AT', and only one repeat per condition are included. Atomic coordinates for MD simulations in this study are available in the PDB database under accession codes 5CKR (MraY) and 1NLM (MurG). Source data are provided with this paper.

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

## Acknowledgements

We thank Drs. Francesco Fiorentino, Tarick El-Baba, Joseph Gault, and Ritu Raj for helpful discussions. We thank Eefjan Breukink (University of Utrecht) for the generous gift of lipid II. Research in the C.V.R. laboratory is supported by a Medical Research Council (MRC) programme grant (MR/V028839/1), on which J.R.B is a Researcher Co-Investigator. J.R.B holds a Royal Society University Research Fellowship and is a Research Fellow at Wolfson College. Research in the WV laboratory is funded by the BBSRC grant BB/R017409/1 (to W.V.). Research in the P.J.S. laboratory is funded by Wellcome (208361/Z/17/Z), the MRC (MR/S009213/1) and BBSRC (BB/P01948X/1, BB/R002517/1 and BB/S003339/1). P.J.S. acknowledges Athena at HPC Midlands, funded by the EPSRC on grant EP/P020232/1, and the University of Warwick Scientific Computing Research Technology Platform for computational access. C.M.B. is funded by an MRC studentship.

## Author contributions

A.O.O., J.R.B., W.V. and C.V.R. designed research; A.O.O., J.R.B. and V.M.H. performed experiments; R.A.C., C.M.B. and P.J.S. performed MD simulations; A.O.O., J.R.B. and C.V.R. wrote the paper with input from all authors.

## Competing interests

C.V.R. is a cofounder of and consultant at OMass Therapeutics. The remaining authors declare no competing interests.
