## [Peer Review File · Nature Communications]

Reviewers' comments:

Reviewer #1 (Remarks to the Author):

The authors demonstrate how native-MS, combined with coarse-grain MD simulation and site directed mutations can be used to assess dimerisation, substrates and antibiotics binding to the PG membrane enzymes MraY, MurG, and MurJ.

Over all I would recommend publication, once my concerns, outlined below, have been addressed.

It's actually correct to annotate charge states as 10+ (for example), and not +10. Ideally it would be annotated z=10+

Figure 1 spectra inset +21: what are the unlabeled additional peaks? Na/NH4 adducts?

To detect the ligand in negative ion, is the instrument polarity switched? Unfortunately, this is not even described in the main or Supporting Information either, but needs to be.

Fig 1f. The isotope ratio/fidelity of the lipid looks very strange. Can the authors please comment? Also, has accurate MS been performed to confirm it is the lipid of interest? And what is the mass accuracy?

Extended figure show x-axis as m/z (3000, 4000 etc) but extended Figure 3 shows m/z x10³. Why this inconsistency in formatting?

Page 6, line 121: "These sites are formed by the Trp-253, Phe-254 122 and Gln-260 on one monomer and Leu-332, Lys-336 and Arg-340 on the other (Fig. 1c)". These sites don't appear to be labelled in Fig 1c.

Page 7, line 127: "hotspots" a little colloquial/non-scientific. Do they mean the two high affinity site in the dimer interface?

Page 12 line 260: why buffer exchanged in to LDAO? The reason is not discussed in the main text of supporting info. therefore a little confusing as to why this addition step as chosen. The only explanation the authors offer is in the final discussion “Our analysis of the oligomeric state of MraY by native mass spectrometry in different detergent micelles shows that MraY is an equilibrium of monomers and dimers, with the latter preferentially binding substrates” which really isn’t that useful.

Page 16, line 348, Final paragraph of the summary: I’d hardly describe this work in the context of "ease of following".

Native MS of membrane proteins and associated complexes is challenging at the best of times. Yes, this is a very powerful approach but by no means routine and available to only a few "skilled labs". I would have liked to see the authors describe a little more how they would integrate this technique with, and how it would further compliments existing analytics. They briefly touch on this in lines 344-346, however, it could be expanded.

Reviewer #2 (Remarks to the Author):

In this manuscript, the authors make use of native mass spec. (MS) and coarse grain (CG) simulation to study components of the peptidoglycan biosynthetic pathway. I was asked to review the computational part. Simulations were performed for two systems, MraY dimer in presence of 2 C55-P molecules per leaflet and MurG monomer in presence of 4 C55-PP molecules per leaflet, both in a POPE lipid bilayer.

1) My main worry is the focus of the manuscript - on subtle details of molecular interactions with lipids in controlling enzyme function and assembly - yet these subtle details seem lacking in the modeling component.

The use of pure POPE lipid bilayers as a proxy for bacterial inner membrane is problematic, as you potentially lose physiologically important lipids interactions with the proteins, which is important in this case, as they will be competing with membrane-bound substrate/product molecules. I do not know offhand what the lipid composition of *Aquifex aeolicus* is, but I would expect to see simulation of “generic” bacterial inner membrane model containing phosphatidylglycerol, phosphatidylethanolamine, cardiolipin, etc, at least.

Also, for MurG, the aim was to gain insights into interaction with lipid I, but simulations were run with C55-PP. It is claimed this is a “reasonable proxy”, but the head group is different, so this does not make much sense. Why not simulate lipid I? It is straightforward to parametrize these molecules, or to obtain parameters from other studies/labs. This shows lack of appropriate rigor in simulation design, given the use of wrong substrate lipid and membrane composition.

2) The simulations do not give much mechanistic insights. CG complexes could be converted to all-atom resolution, where simulations can give information on stability, key interactions, dynamics induced by the ligands. It would also be insightful to look at energetics of monomer-dimer equilibria of MraY in the presence of different ligands with long timescale or biased molecular dynamics.

3) MraY simulations indicated possible binding sites for C55-P on the dimer surface. Subsequent mutation of some of these residues and native MS then confirmed the dimer was destabilized for some (particularly R340). If I understand correctly, other point mutants did not disrupt the substrate-induced monomer/dimer population. Why? Perhaps atomistic simulations would help?

4) Where did the choice of number of C55-P molecules per leaflet come from? Was this arbitrary?

5) There is no statistical analysis, no evidence of convergence in observed binding properties, etc.

6) Some basic analysis to confirm stability of simulation systems, protein dynamics, etc is missing.

7) No explanation is given for how protein secondary/tertiary structure is modeled - this needs to be clarified. What about the MraY dimer - were individual chains free to move, and if so, did ligand-free protein disassemble into monomeric states?

Reviewer #3 (Remarks to the Author):

Oluwole et al. attempted to probe substrate and inhibitor binding to MraY, MurG, and MurJ using native mass spec analysis; these integral membrane protein enzymes are essential for peptidoglycan biosynthesis in bacteria. The major findings of this study were the following:

1) Monomer-dimer equilibrium and enzyme function of MraY is regulated by the isoprenoid lipid undecaprenyl phosphate (C55-P) – a key 55-long carbon chain lipid required for the synthesis of the bacterial cell wall. They found that C55-P binds at the dimer interface of MraY.

2) MurG and MurJ have apparent increased affinities towards their respective lipid substrates and these affinity gradients appear to be important for peptidoglycan biosynthesis.

The use of native mass spec for monitoring the initial steps of the lipid cycle of the peptidoglycan synthesis is interesting. However, I have several serious concerns regarding the authors' main conclusions drawn from their experiments, which are not substantiated by the experimental data nor consistent with previous studies on MraY.

Major comments

1) The author observed that C55-P is bound to MraY by native mass spec. Previous crystallographic studies found electron density at the dimer interface of MraY that could be either C55-P or lipid. On the basis of MD simulation and mutagenesis studies, the authors concluded that C55-P is bound to the dimer interface with a 2:2 or 2:1 ratio. The existence of C55-P at the dimer interface and its effect on the monomer-dimer equilibrium in detergent micelles is convincing. However, what is concerning is the authors' claim that this C55-P is involved in catalysis of lipid I formation and the implication that the dimer interface is where catalysis takes place in MraY. The authors further suggest that lipid I is formed at the dimer interface from the catalytic activity of MraY (Figure 6). Authors' mass spec and functional data do not support this claim. Furthermore, this claim is contradictory to the rigorous, long-standing biochemical and structural data about MraY. We know from these previous studies that 1) The active site where UDP-MurNAc-pentapeptide and C55-P bind to and are catalyzed to form lipid I is well-established by decades of biochemical and structural studies of MraY. This active site is away from the dimer interface, and makes it structurally impossible for the C55-P in the dimer interface to reach the active site for catalysis. It is well established that C55-P has to attack UDP-MurNAc-pentapeptide for lipid I formation but it is impossible for the C55-P at the dimer interface to reach UDP-MurNAc-pentapeptide in the active site. 2) It is well known that many natural product inhibitors such as mureidomycin and liposidomycin compete with C55-P for MraY binding. A recent crystal structure of MraY in complex with a liposidomycin analog shows the acyl chain of the liposidomycin analog, which mimics the lipid tail of C55-P, bound to the active site (near TM9 and TM5), strongly suggesting that this is the C55-P binding site for lipid I formation, which is away from the dimer interface. 3) Also, the crystal structure of MraY-tunicamycin complex shows that the tunicamycin binding site is at the active site, and not at the dimer interface. However, the authors claim that tunicamycin inhibits MraY by competing with C55-P binding at the dimer interface. Because the author's proposal of the role of C55-P at the dimer interface is contradictory to all MraY studies, authors must provide substantiated biochemical and functional data to prove that the C55-P in the dimer interface is involved in catalysis. In my scientific view, the C55-P at the dimer interface could have either a regulatory or structural role.

2) Based on the above-mentioned reasoning, MraY must bind to C55-P for Lipid I formation in addition to the C55-P in the dimer interface. The author suggests that MraY:C55-P stoichiometry is either 2:2 or 2:1. However, in the extended data figures 6 and 7, exogenous C55-P in the titration experiment showed that monomeric MraY can bind to two C55-P, suggesting that the stoichiometry can be up to 2:4. It is possible that the catalytic C55-P is not stably bound to MraY and thus removed during the course of purification in the presence of detergent micelles. Authors must perform titration experiments with a very high concentration of exogenous C55-P (> 20 μ M C55-P) using a mild detergent to preserve dimeric MraY to see if the stoichiometry can go up to 2:4.

3) In Extended Data Fig. 6, Titration of wild type (WT) MraY and Q260A/R340A mutant (mt) with exogenous C55-P revealed that the WT MraY shows a higher affinity for C55-P. I am perplexed by this data as to why the authors used the harsher LDAO detergent to make monomeric MraY, which seems contradictory to the finding that dimeric MraY is necessary for C55-P binding, or does the C55 binding influence the dimer formation? The authors should perform the experimental condition similar to Fig.1b where inclusion of C55-P increases the dimer fraction.

4) Fig. 1b, Fig. 2b, Extended Data Fig. 6c. Information is missing such as the number of experimental measurements; whether these measurements are technical or biological replicates; and whether they are statistically significant. I strongly suggest that the measurements be taken as biological replicates, as the difference between WT and mt MraY are subtle.

5) Although monitoring the coupled reactions of MraY-MurG-MurJ by native mass spec is a unique approach, the interpretation from experimental data that the affinity for lipid I and lipid II increases in MurG and MurJ is questionable. In the coupled synthesis scheme, it is expected that enzymes bind their products with lower affinity than their substrates in order to facilitate intermediate exchange within the pathway. The apparent lower affinity for product is effectively lower because the incoming substrate will compete for the active site so the product release will be facilitated. To make the claim that differential affinities play a role in MraY-MurG-MurJ, the authors should measure their relative affinities for lipid I and lipid II in isolated systems instead of the coupled system, similar to what they did with C55-P.

6) Throughout the figures, authors place lipid I at the dimer interface; as if it is in the active site. The lipid I binding at the dimer interface has not been demonstrated experimentally. Neither do the MD simulations nor mutagenesis studies show that lipid I binds to the dimer interface. The author should revise their figures and this claim.

7) Throughout the text, it is very confusing to the reader when the authors do not distinguish between C55-P binding from “substrate” binding. For example, “Overall, these data experimentally confirm predictions of substrate binding residues from MD simulations and highlight in particular the significant role of R340 in substrate binding.” Authors should avoid using “substrate” for C55-P at the dimer interface. UDP-MurNac-pentapeptide is also a substrate, and the C55-P at the dimer interface is not shown to be involved in lipid I formation.

Reviewer #1 (Remarks to the Author):

The authors demonstrate how native-MS, combined with coarse-grain MD simulation and site directed mutations can be used to assess dimerisation, substrates and antibiotics binding to the PG membrane enzymes *MraY*, *MurG*, and *MurJ*. Over all I would recommend publication, once my concerns, outlined below, have been addressed.

We thank the reviewer for their positive response and strong support of our work. All the concerns have now been addressed in our revised version. Please see below for our point by point response.

It's actually correct to annotate charge states as 10+ (for example), and not +10. Ideally it would be annotated as $z=10+$

Thank you for this suggestion. This is now fixed throughout.

Figure 1 spectra inset +21: what are the unlabeled additional peaks? Na/NH₄ adducts?

These low intensity peaks based on their mass differences are too large to assign to Na⁺ or NH₄⁺ adducts. Based on the observed masses we assign these additional peaks as C₅₅-PP and various forms of lipid I and lipid II (see below RF1). Masses of these adducts, and other species, are now included in Supplementary Table 1.

*RF1: Expansion of the 21+ charge state from the spectrum shown in Fig 1a. All of these adducts are bound to dimeric *MraY*.*

To detect the ligand in negative ion, is the instrument polarity switched? Unfortunately, this is not even described in the main or Supporting Information either, but needs to be.

*We apologise for this omission. We switched the instrument into negative polarity and tuned the ion optics accordingly to detect C₅₅-P and lipid I released from *MraY* dimer. This information is now explicitly stated in the supporting information under native mass spectrometry section (page 7) as follows:*

“Unless otherwise stated all the experiments were performed in the positive polarity. For the CID MS/MS experiments to detect the ligands bound to proteins, negative polarity was used and the instrument ion optics were tuned accordingly to enhance transmission and detection of low m/z ions”.

Fig 1f. The isotope ratio/fidelity of the lipid looks very strange. Can the authors please comment?

We thank the reviewer for this observation. We have repeated the analysis of the lipid and provide new spectra. Of note, the isotopic distribution in the spectra matches closely the calculated isotopic ratios (now included as inserts in Extended Data Fig 1c and 1f).

Also, has accurate MS been performed to confirm it is the lipid of interest? And what is the mass accuracy?

Yes, the observed masses for C₅₅-P (847.9±1.0 in the context of MraY dimer; 846.6627 Da when dissociated) and for lipid I (1716.7±2.3 Da in the context of MraY dimer; 1715.9749 Da when dissociated). These masses are in excellent agreement with the calculated monoisotopic masses of 846.66495 Da (for C₅₅-P) and 1715.9627 Da (for lipid I). These masses are now included in the main text on page 4 as follows:

“In addition, we observed low intensity peaks that can be assigned to undecaprenyl diphosphate (926.24±2.70 Da), cardiolipin (1402.90±1.83 Da), and lipid II (1920.09±2.11 Da) copurifying with dimer MraY (Supplementary Table S1).”

To reflect fidelity of the observed isotopic distributions and mass accuracies, in the legend to Extended Data Fig.1, we include:

“Theoretical isotopic distributions of C₅₅-P and lipid I shown as inserts in panel c and f are calculated using an online tool (<https://prospector.ucsf.edu/prospector/cgi-bin/msform.cgi?form=msisotope>). The observed isotopic pattern closely matches the theoretical isotopic distributions. For C₅₅-P [(C₅₅H₉₀O₄P)⁻], the observed m/z = 845.6627 agrees well with the calculated monoisotopic m/z = 845.6571 (mass deviation of 6.6 ppm). For the endogenous lipid I [(C₈₇H₁₄₂N₇O₂₃P₂)⁻], the observed m/z = 1714.9749 Da while the calculated monoisotopic m/z = 1714.9627 (mass deviation of 7.1 ppm).”

Extended figure show x-axis as m/z (3000, 4000 etc) but extended Figure 3 shows m/z x10³. Why this inconsistency in formatting?

This has been fixed with m/z scales now in both cases.

Page 6, line 121: “These sites are formed by the Trp-253, Phe-254 122 and Gln-260 on one monomer and Leu-332, Lys-336 and Arg-340 on the other (Fig. 1c)”. These sites don’t appear to be labelled in Fig 1c.

We thank the reviewer for this suggestion. We have labelled these residues in the new data set with the atomistic MD simulation in Fig. 2c and in Extended Data Fig. 4a.

Page 7, line 127: “hotspots” a little colloquial/non-scientific. Do they mean the two high affinity site in the dimer interface?

We agree, the sentence is now rephrased as follows:

“Guided by the high affinity C₅₅-P binding sites predicted by the MD simulation results”.

Page 12 line 260: why buffer exchanged in to LDAO? The reason is not discussed in the

main text of supporting info. therefore a little confusing as to why this addition step as chosen. The only explanation the authors offer is in the final discussion “Our analysis of the oligomeric state of MraY by native mass spectrometry in different detergent micelles shows that MraY is an equilibrium of monomers and dimers, with the latter preferentially binding substrates” which really isn’t that useful.

We have justified the additional buffer exchange step in the Supplementary Methods (page 6) as follows:

“For the *in situ* concerted reactions, the proteins were buffer-exchanged into LDAO as it provides sufficient stability for all three enzymes MraY, MurG and MurJ. Of note, detergents used to study MraY (C8E4 and OG) caused precipitation of MurG and MurJ.

Page 16, line 348, Final paragraph of the summary: I’d hardly describe this work in the context of “ease of following”.

Native MS of membrane proteins and associated complexes is challenging at the best of times.

We thank the reviewer for their appreciation of the technical challenges involved in performing native MS experiments.

We have now re-phrased the sentence as follows:

“By studying three membrane enzymes with multiple substrates, we demonstrated the potential for following biosynthetic reactions using native MS and reveal the relative binding affinities that may drive precursor passage during peptidoglycan synthesis.”

Yes, this is a very powerful approach but by no means routine and available to only a few “skilled labs”. I would have liked to see the authors describe a little more how they would integrate this technique with, and how it would further compliments existing analytics. They briefly touch on this in lines 344-346, however, it could be expanded.

We have expanded further on how native MS compliments existing methodologies on page 22-23, we include:

“From a methodological standpoint, results from this study highlight how native MS, in conjunction with MD and structural information, can complement other biophysical approaches. By defining relative binding affinities for different lipid substrates and products, native MS can inform high resolution structures wherein density for flexible hydrophobic ligands can be difficult to define.”

Reviewer #2 (Remarks to the Author):

In this manuscript, the authors make use of native mass spec. (MS) and coarse grain (CG) simulation to study components of the peptidoglycan biosynthetic pathway. I was asked to review the computational part. Simulations were performed for two systems, MraY dimer in presence of 2 C55-P molecules per leaflet and MurG monomer in presence of 4 C55-PP molecules per leaflet, both in a POPE lipid bilayer.

1) My main worry is the focus of the manuscript - on subtle details of molecular interactions with lipids in controlling enzyme function and assembly - yet these subtle details seem lacking in the modeling component.

We thank the referee for their careful reading of the manuscript and constructive comments. We have now made extensive additions to the MD data (Extended Data Figs. 4 and 9). Specifically, we have included 450 μ s of new CG MD data and ca. 2 μ s of atomistic MD data, along with many additional analyses and figure panels (Fig. 1c, Fig. 4c, d, Extended Data Figs. 4 and 9).

The use of pure POPE lipid bilayers as a proxy for bacterial inner membrane is problematic, as you potentially lose physiologically important lipid interactions with the proteins, which is important in this case, as they will be competing with membrane-bound substrate/product molecules. I do not know offhand what the lipid composition of *Aquifex aeolicus* is, but I would expect to see simulation of “generic” bacterial inner membrane model containing phosphatidylglycerol, phosphatidylethanolamine, cardiolipin, etc, at least.

As this work focuses on interactions with specific lipid substrates we initially chose to use POPE bilayers, as simple but effective model systems, as is standard in the field. However, the referee raises a good point about potential competition of anionic lipids with the substrate. We have therefore now performed all CG data using a mixed membrane composition of ~ 7:3:1 POPE: POPG: cardiolipin, along with the substrate lipid. Although the data is largely unaffected by the presence of the anionic lipids we agree that this addition makes our dataset more compelling.

Accordingly, in the Methods section (page 8), we have included:

“For MraY, the protein was then built into a mixed membrane with 65% palmitoyl-2-oleoyl-*sn*-glycero-3-phosphoethanolamine (POPE), 24% palmitoyl-2-oleoyl-*sn*-glycero-3-phosphoglycerol (POPG), 10% cardiolipin using the *insane* protocol⁷, with ca. 1% of either C₅₅-P (in a charge state $z = -1$) or lipid I. For MurG, the protein was randomly oriented in 3D space and then a membrane was built 5 nm from the protein. The lipid composition of the model membrane was as for MraY, but with either 1% lipid I or 1% lipid II. The concentration of substrate/product was chosen to match the biological concentration of lipid II of about 1% (ref.8)

Also, for MurG, the aim was to gain insights into interaction with lipid I, but simulations were run with C₅₅-PP. It is claimed this is a “reasonable proxy”, but the head group is different, so this does not make much sense. Why not simulate lipid I? It is straightforward to parametrize these molecules, or to obtain parameters from other studies/labs. This shows lack of appropriate rigor in simulation design, given the use of wrong substrate lipid and membrane composition.

We previously selected C₅₅-PP as a proxy because we expected the data to be qualitatively similar to that of lipid I binding reasoning that this would be sufficient to complement the MS data. However, we accept that this does not include the lipid I headgroup. We have therefore now obtained CG parameters for both lipid I and II and repeated all MurG data with lipid I and lipid II present (Fig. 4c,d; Extended Data Fig. 9). Inclusion of these lipids provides insight into how the lipid I and II headgroups interact with MurG and we thank the reviewer for encouraging us to do this.

For MurG interactions of lipid I and lipid II, we have now included:

“We initiated simulations with monomeric MurG positioned ca. 10 nm from a model membrane containing either lipid I or lipid II. We then ran the simulations for 15 μ s to allow MurG to bind to the membrane. Analysis of the binding data with PyLipID reveals three prominent binding sites for lipid I and lipid II on MurG: two of these

sites are in close proximity, centred on the helix from Lys-72 to Lys-93 (Fig. 4c), and were previously predicted to be important for MurG interaction with the membrane.¹⁴ A third site also exists, nearer the structurally-resolved substrate binding site.²⁹ (Fig. 4c; magenta). This site involves the peptide and sugar headgroups of lipid I and lipid II making extensive contacts with the entire substrate-binding region (Extended Data Fig. 9). Interestingly, one of the primary lipid II binding poses involves the GlcNAc of lipid II interacting with several of the GlcNAc-coordinating residues in the cocrystal structure of MurG²⁹, these residues include Phe-21, Ser-192 and Leu-265 (Fig. 4d). We then converted a bound CG pose to atomistic and performed a set of MD simulations, which further supported the stability of this binding orientation (Extended Data Fig. 9). This pose might represent an enzymatic intermediate state, whereby GlcNAc has just been added to lipid I to form lipid II, which will now be released from the active site. Across the CG MD data, we observed on average 2.15 ± 0.5 lipid I molecules bound with high affinity to MurG sites, in reasonable accord with the MS data. Analysis of k_{off} from these simulations reveals that lipid II is less strongly bound to MurG than lipid I by a factor of about 2-3 ($k_{\text{off}} = 0.70 \pm 1.03 \mu\text{s}^{-1}$ for lipid I and $k_{\text{off}} = 1.91 \pm 0.30 \mu\text{s}^{-1}$ for lipid II) at this site. The MD results are consistent with the apparent K_{d} determined by native MS experiments above.”

In addition, we have also added analysis of MraY interacting with lipid I (Extended data Figure 4c). We found that lipid I also interacts with MraY via the carrier lipid hydrophobic tail near the dimer interface, and on page 7, we have now included:

“These sites have very similar occupancies in the lipid I dataset (see Extended Data Fig. 4c for the equivalent pose), suggesting that both C₅₅-P and lipid I interact considerably with MraY through this interfacial region.”

2) The simulations do not give much mechanistic insights. CG complexes could be converted to all-atom resolution, where simulations can give information on stability, key interactions, dynamics induced by the ligands. It would also be insightful to look at energetics of monomer-dimer equilibria of MraY in the presence of different ligands with long timescale or biased molecular dynamics.

This is a good suggestion. We have now included atomistic data for MraY with C₅₅-P (Fig. 1c), and MurG with Lipid II (Fig.4d). In both cases we have used the data to infer more information on how these lipids interact with the protein. In the case of MraY, we have now included RMSDs for the dimer with and without C₅₅-P present (Extended Data Fig. 4d). The energetics of MraY monomer-dimer equilibria would be interesting to study, but unfortunately this is not readily achievable with MD, even when using CG.

3) MraY simulations indicated possible binding sites for C₅₅-P the dimer surface. Subsequent mutation of some of these residues and native MS then confirmed the dimer was destabilized for some (particularly R340). If I understand correctly, other point mutants did not disrupt the substrate-induced monomer/dimer population. Why? Perhaps atomistic simulations would help?

Yes, atomistic simulations were helpful here. R340 was the residue which interacted the most strongly with C₅₅-P. We found that C₅₅-P molecule bound at the interfacial sites make extensive contact with both protomers via the hydrophobic tail but the phosphate head will

*bridge the two *MraY* protomers, this explain why mutations of complementary residues on both protomers are required to substantially disrupt the binding at the interface, and consequently yield monomeric *MraY*. We have now highlighted this point on page 8 by including:*

“Although R340 residue has the lowest k_{off} from the MD simulation results, mutating this single residue was found insufficient to completely disrupt C_{55} -P binding to *MraY* at the interface. This can be attributed to extensive contacts made by the lipid substrate to both *MraY* protomers when bound at the interface (Fig. 2c). However, a substantial disruption of the interfacial sites was achieved by mutating the complementary pair of residues on each *MraY* protomer as observed in the case of Q260A/R340A and W253A/F254A/R340A, and consequently these *MraY* mutants are predominantly monomeric.”

4) Where did the choice of number of C_{55} -P molecules per leaflet come from? Was this arbitrary?

Great question. This was based on estimated concentrations of the substrate molecules in the membrane. This is an important point and we have now added an explanation and reference for this to the Methods (Pages 8 and 9) as follows:

“For MurG, the protein was randomly oriented in 3D space and then a membrane was built 5 nm from the protein. The lipid composition of the model membrane was as for *MraY*, but with either 1% lipid I or 1% lipid II. The concentration of substrate/product was chosen to match the biological concentration of lipid II of about 1% (ref.8)”

5) There is no statistical analysis, no evidence of convergence in observed binding properties, etc.

We apologise for this omission. For the lipid binding data, the data are bootstrapped to provide convergence and error estimates for the k_{off} . This is the metric used predominantly here. We have now reported bootstrapped values for the identified sites (Extended Data Fig. 4d).

6) Some basic analysis to confirm stability of simulation systems, protein dynamics, etc is missing.

This type of analysis is not usually required for CG simulations, where the protein structure is restrained by an elastic network. As we have now added atomistic data and included the RMSD values accordingly (Extended data Figs. 4d, 9d).

7) No explanation is given for how protein secondary/tertiary structure is modeled - this needs to be clarified.

We have added more details to the Methods section (Page8) as follows:

“Coarse grained simulations were built using the structure of *MraY* in a dimer state (PDBs code: 5CKR) or chain B of MurG (PDB code: 1NLM). Selected poses from the CG data which matched the optimal binding modes calculated by PyLipID were then converted to an atomistic description using CG2AT2.”

What about the *MraY* dimer - were individual chains free to move, and if so, did ligand-free protein disassemble into monomeric states?

We thank the reviewer for helpful comments. How C_{55} -P mediates the monomer-dimer assembly was investigated experimentally and not via the MD-simulations. This is because we aimed at identifying the key residues that mediate high-affinity binding of C_{55} -P molecules

at the interfacial sites. Elastic networks were applied to prevent the dimer *MraY* from collapsing. We have provided this information in the Method section (Supplementary, page 9):

“The proteins were converted to Martini 2.2 using the *martinize* method, and 1,000 kJ mol⁻¹ nm⁻² elastic networks were applied between backbone beads within 1 nm. This helped stabilise the secondary and tertiary structures of each system, and in the case of *MraY*, stabilise the dimer and stop the dimer interface from collapsing.”

Reviewer #3 (Remarks to the Author):

Oluwole et al. attempted to probe substrate and inhibitor binding to *MraY*, *MurG*, and *MurJ* using native mass spec analysis; these integral membrane protein enzymes are essential for peptidoglycan biosynthesis in bacteria. The major findings of this study were the following: 1) Monomer-dimer equilibrium and enzyme function of *MraY* is regulated by the isoprenoid lipid undecaprenyl phosphate (C55-P) – a key 55-long carbon chain lipid required for the synthesis of the bacterial cell wall. They found that C55-P binds at the dimer interface of *MraY*. 2) *MurG* and *MurJ* have apparent increased affinities towards their respective lipid substrates and these affinity gradients appear to be important for peptidoglycan biosynthesis.

The use of native mass spec for monitoring the initial steps of the lipid cycle of the peptidoglycan synthesis is interesting. However, I have several serious concerns regarding the authors' main conclusions drawn from their experiments, which are not substantiated by the experimental data nor consistent with previous studies on *MraY*.

We thank the reviewer for highlighting the main findings of our study. The potential issues of concerns have been fully addressed in the new version of the manuscript. We have carried out many additional experiments and provide new data and analysis (Fig. 2d, Fig.4, Extended Data Figs. 7a, 8-10). We have also improved presentation of our findings to ensure clarity. Details are provided in response to individual points below:

Major comments

1) The author observed that C55-P is bound to *MraY* by native mass spec. Previous crystallographic studies found electron density at the dimer interface of *MraY* that could be either C55-P or lipid. On the basis of MD simulation and mutagenesis studies, the authors concluded that C55-P is bound to the dimer interface with a 2:2 or 2:1 ratio. The existence of C55-P at the dimer interface and its effect on the monomer-dimer equilibrium in detergent micelles is convincing.

We thank the reviewer for a careful reading of the manuscript, and many useful suggestions and constructive comments.

However, what is concerning is the authors' claim that this C55-P is involved in catalysis of lipid I formation and the implication that the dimer interface is where catalysis takes place in *MraY*. The authors further suggest that lipid I is formed at the dimer interface from the catalytic activity of *MraY* (Figure 6). Authors' mass spec and functional data do not support this claim. Furthermore, this claim is contradictory to the rigorous, long-standing biochemical and structural data about *MraY*. We know from these previous studies that

1) The active site where UDP-MurNAc-pentapeptide and C55-P bind to and are catalyzed to form lipid I is well-established by decades of biochemical and structural studies of *MraY*. This active site is away from the dimer interface, and makes it structurally impossible for the C55-P in the dimer interface to reach the active site for catalysis. It is well established that C55-P has to attack UDP-MurNAc-pentapeptide for lipid I formation but it is impossible for the C55-P at the dimer interface to reach UDP-MurNAc-pentapeptide in the active site.

*We apologise for any confusion. We acknowledge the excellent decades of biochemical and structural studies on *MraY* which establish that *MraY*-mediated catalysis takes place at the catalytic cleft comprising the HHH motif and the aspartate residues (Asp117, 118 and 265 in *A. aeolicus MraY*). We agree with the reviewer that catalysis does not occur at the dimer interface. Here we ascribe a dimer-stabilising role to the C₅₅-P molecules found to bind at the dimer interface of *MraY*.*

To ensure clarity of where lipid I formation occurs, we have revised the cartoon depictions in all applicable figures and re-phrased relevant text. For example (page 11):

“Although *MraY* catalysis does not occur at the dimer interface, molecules of lipid I synthesized *in situ* were captured as adducts to the dimer (mass 1674 Da).”

And on page 20:

“The catalytic aspartate residues on each *MraY* protomer (ref. 17,36,37) are remote from the dimer interface and distinct from the C₅₅-P coordinating residue R340 reported in this study.”

The structural role of interfacial lipids identified in this study is now reflected more clearly throughout the manuscript, for example on page 20 we include the following:

“We therefore ascribe structural roles to the interfacial lipids found to bind *MraY* dimer”.

“Given the observed coupling between dimerization and the interfacial binding of endogenous C₅₅-P and lipid I in our mutagenesis results, the interfacial ligands must correspond to structural lipids that provide stability for the *MraY* dimer.”

2) It is well known that many natural product inhibitors such as mureidomycin and liposidomycin compete with C55-P for *MraY* binding. A recent crystal structure of *MraY* in complex with a liposidomycin analog shows the acyl chain of the liposidomycin analog, which mimics the lipid tail of C55-P, bound to the active site (near TM9 and TM5), strongly suggesting that this is the C55-P binding site for lipid I formation, which is away from the dimer interface.

*We acknowledge the excellent work of Hakulinen et al Nat Chem Biol 13, 265 (2017) and other studies by S-Y Lee's group (notably Nat. Commun. 10, 2917 (2019)) in this regard. We agree that the catalytic residues are away from the dimer interface of *MraY* and acknowledge that *MraY* is able to bind more than one molecule of C₅₅-P per protomer (Fig. 2d, Extended Data Figs. 3 and 6). We now reiterate this as follows:*

And on page 21:

“This is consistent with our MD simulation data that reveal relatively weaker C₅₅-P binding near the proposed active site than at the interfacial sites. While this indicates that *MraY* can interact with C₅₅-P at multiple sites with different affinities, it raises a question regarding the role of C₅₅-P molecules whose phosphate groups bind adjacent to the dimer interface. Given the observed connection between dimerization and the interfacial binding of endogenous C₅₅-P and lipid I, binding of these molecules at the interfacial sites likely provides structural

stability to MraY dimer. We therefore ascribe structural roles to the interfacial lipids bound to MraY dimer at the interfacial sites.”

To describe ligand bound at the dimer interface, we now qualify with the word “interfacial”.

For instance, on page 8, we now have:

“Together these results show that disruption of interfacial C₅₅-P binding residues is coupled with destabilization of the MraY dimer.

3) Also, the crystal structure of MraY-tunicamycin complex shows that the tunicamycin binding site is at the active site, and not at the dimer interface. However, the authors claim that tunicamycin inhibits MraY by competing with C55-P binding at the dimer interface.

We have removed the statement leading to a misunderstanding that tunicamycin inhibits MraY at the dimer interface. This is now replaced with:

“Although tunicamycin competes with the binding of UM5 at the catalytic site,²⁵⁻²⁷ our data show that the hydrophobic tails of tunicamycin can also stabilise the MraY dimer (Fig. 1b).”

Because the author’s proposal of the role of C55-P at the dimer interface is contradictory to all MraY studies, authors must provide substantiated biochemical and functional data to prove that the C55-P in the dimer interface is involved in catalysis. In my scientific view, the C55-P at the dimer interface could have either a regulatory or structural role.

We thank the reviewer for sharing their expertise. We agree that interfacial C₅₅-P molecules likely provide a structural role as evident in other membrane protein oligomers {*Nature* **541**, 421-424 (2017); *Cell Chem. Biol.* **25**, 840-848 e4 (2018)}. Consistent with this idea, DMPG has been previously reported to enhance dimerization and enzymatic activities of purified MraY {*Elife* **6**, e20954 (2017)}.

2) Based on the above-mentioned reasoning, MraY must bind to C55-P for Lipid I formation in addition to the C55-P in the dimer interface. The author suggests that MraY:C55-P stoichiometry is either 2:2 or 2:1. However, in the extended data figures 6 and 7, exogenous C55-P in the titration experiment showed that monomeric MraY can bind to two C55-P, suggesting that the stoichiometry can be up to 2:4. It is possible that the catalytic C55-P is not stably bound to MraY and thus removed during the course of purification in the presence of detergent micelles. Authors must perform titration experiments with a very high concentration of exogenous C55-P (> 20 μ M C55-P) using a mild detergent to preserve dimeric MraY to see if the stoichiometry can go up to 2:4.

Yes, we agree that MraY can bind multiple C₅₅-P molecules both at the active site and at the dimer interface. As suggested, we have performed the experiment with 5 μ M MraY and 0-50 μ M C₅₅-P. We find that MraY dimer can bind multiple C55P molecules, up to 4 bound adducts are seen (see Figure R2 below). This data is now included in Extended Fig 3.

RF2: Native mass spectrum of 5 μ M MraY and 50 μ M C₅₅-P released from C8E4 micelles. The mass of 2xC₅₅-P = the mass of 1x lipid I.

To clarify further, we have re-phrased in the discussions section as follows:

“Briefly, our MS data show that MraY can exist in a monomer-dimer equilibrium and that the dimers bind C₅₅-P and/or lipid I more efficiently than the monomers, with protein: lipid binding stoichiometry of ~2:4.”

3) In Extended Data Fig. 6, Titration of wild type (WT) MraY and Q260A/R340A mutant (mt) with exogenous C₅₅-P revealed that the WT MraY shows a higher affinity for C₅₅-P. I am perplexed by this data as to why the authors used the harsher LDAO detergent to make monomeric MraY, which seems contradictory to the finding that dimeric MraY is necessary for C₅₅-P binding, or does the C₅₅ binding influence the dimer formation? The authors should perform the experimental condition similar to Fig.1b where inclusion of C₅₅-P increases the dimer fraction.

Yes, we did find that C₅₅-P affects dimer formation and that little or no dimer formation is observed with the Q260A/R340A mutant. We initially selected LDAO (a harsh delipidating detergent) to compare the relative binding affinity of the wild-type and mutants in order to remove endogenous ligands copurified with the wild type. As requested, we have now carried out a repeat experiment using the conditions used to record the data in Fig. 1b. Overall the two sets of conditions yielded similar conclusions that the wild type can bind C₅₅-P more efficiently than the mutant. The new result is shown (Fig. 2d) and explained as follows:

“We next tested whether exogeneous C₅₅-P can induce dimerization of wild-type MraY and the Q260A/R340A mutant by incubating the protein variants with a 10-fold molar excess of C₅₅-P and recorded the spectra by releasing the proteins from C8E4 micelles. The resulting spectra show that the wild-type MraY binds to C₅₅-P and that the relative proportions of dimers were enhanced. In contrast, the Q260A/R340A MraY mutant failed to dimerise and no C₅₅-P binding was observed under the same conditions used for the wild-type (Fig. 2d).”

4) Fig. 1b, Fig. 2b, Extended Data Fig. 6c. Information is missing such as the number of experimental measurements; whether these measurements are technical or biological replicates; and whether they are statistically significant. I strongly suggest that the

measurements be taken as biological replicates, as the difference between WT and mt MraY are subtle.

We thank the reviewer for this helpful suggestion. We have included number of replications where applicable. Of note, we performed mutagenesis experiments as biological replicates. This point is now clearly stated in the legend of Fig. 2a (page 9) as follows:

“Each protein was expressed and purified at least 3 times starting with a fresh transformation. Error bars are standard error calculated from 3 replicate measurements.”

and in the Supplementary methods (Page 5), we include:

“All MraY variants (wild type and mutants) were expressed and purified at least 3 times under the same conditions, starting with single colonies from a freshly transformed C41(DE3) host strain.”

We have plotted standard deviations of 3 replicate measurements wherever average intensities are reported. In Fig.1b, Fig. 2b, Extended Data Fig.6c, we have included this information as appropriate:

“Error bars are standard error calculated from 3 replicate measurements.”

5) Although monitoring the coupled reactions of MraY-MurG-MurJ by native mass spec is a unique approach, the interpretation from experimental data that the affinity for lipid I and lipid II increases in MurG and MurJ is questionable. In the coupled synthesis scheme, it is expected that enzymes bind their products with lower affinity than their substrates in order to facilitate intermediate exchange within the pathway. The apparent lower affinity for product is effectively lower because the incoming substrate will compete for the active site so the produce release will be facilitated. To make the claim that differential affinities play a role in MraY-MurG-MurJ, the authors should measure their relative affinities for lipid I and lipid II in isolated systems instead of the coupled system, similar to what they did with C55-P.

We thank the reviewer for their comments and this idea. We originally set out to perform a coupled synthesis reaction (rather than pair-wise binding) because in the native membrane, these enzymes do not work in isolation. We therefore consider our methodological capability to directly study these biosynthetic enzymes in a “coupled system” by native MS as a unique advantage.

However, as suggested, we have now implemented the idea in two ways:

(i) We performed systematic ligand binding experiments with MurG, and obtained K_d s for MurG-lipid I and MurG-lipid II interactions (Fig.4a,b and Extended data Fig.8). We explain this data by including:

“To investigate affinity of MurG for its native substrate lipid I, we recorded spectra for solutions containing 5 μ M MurG and 0-25 μ M lipid I (Fig. 4a). At lipid I concentration of 1.5 μ M, charge state series with an additional mass of 1717 Da, consistent with one molecule of lipid I bound to MurG, were observed. Further increase in lipid I concentration yielded additional charge state series reflecting binding of second lipid I molecule to MurG. Notably at MurG: lipid I molar ratios ≥ 1 , the MurG-lipid I complex becomes predominant over the unbound form of MurG. A fit to the relative intensity of lipid I-bound MurG at different concentrations for the first lipid I binding event yielded an apparent dissociation constant (K_d) of $1.9 \pm 0.4 \mu$ M and a Hill coefficient of $h = 2.6 \pm 1.2$ (Fig. 4b). This indicated that lipid I molecules bind to MurG with a high affinity and at more than one site with positive cooperativity. Equivalent analysis of MurG-lipid II binding interactions yielded a higher apparent K_d of $2.6 \pm 0.9 \mu$ M (Fig. 4b, Extended Data Fig. 8) than for lipid I. Overall, these

data indicate that MurG binds its native substrate lipid I with a higher affinity than its glycosylated product lipid II.”

(ii) We performed pairwise comparisons of binding affinities of MraY, MurG, and MurJ with their respective substrate and product. The results are now included as the new Extended Data Fig.10. Overall, the new data set validate the results of our coupled reaction experiments in that each enzyme binds with higher affinity to its own physiological lipid precursor than to its product.

We described the results by adding:

“To validate these results, we assessed the relative affinities of MraY, MurG and MurJ for their respective lipid substrates individually and compared the products in a pair-wise manner. To enable direct comparison, the membrane enzymes were prepared in the same buffer (0.5% LDAO, 200 mM ammonium acetates, pH 8.0) and spectra were recorded using the same instrument settings (activation of 100 V). The resulting data for MraY and MurG show that C₅₅-P binds more favourably to MraY (Extended Data Fig. 10a, 10b). This is in accordance with the fact that C₅₅-P is the native substrate of MraY. Similarly, we find that lipid I binds to MurG with higher intensity than to MraY (Extended Data Fig. 10c, 10d), indicating that an MraY protomer readily releases lipid I for further processing by MurG. Analogously lipid II binds to the flippase MurJ with a higher affinity than to MurG, the preceding enzyme in the pathway (Extended Data Fig. 10e, 10f). These observations are consistent with the binding gradients observed in our *in-situ* competitive assay mixtures (Fig. 5).”

6) Throughout the figures, authors place lipid I at the dimer interface; as if it is in the active site. The lipid I binding at the dimer interface has not been demonstrated experimentally. Neither do the MD simulations nor mutagenesis studies show that lipid I binds to the dimer interface. The author should revise their figures and this claim.

The new set of data from MD simulations (Extended data Fig. 4c) supported our depiction of lipid I interacting with the MraY dimer considerably at the interfacial region. However, we agree that can be confusing and, as suggested, we have modified the illustrations on relevant figures to show that lipid I can bind MraY also at the active site.

7) Throughout the text, it is very confusing to the reader when the authors do not distinguish between C55-P binding from “substrate” binding. For example, “Overall, these data experimentally confirm predictions of substrate binding residues from MD simulations and highlight in particular the significant role of R340 in substrate binding.” Authors should avoid using “substrate” for C55-P at the dimer interface. UDP-MurNac-pentapeptide is also a substrate, and the C55-P at the dimer interface is not shown to be involved in lipid I formation.

We thank the reviewer for their expertise and helpful comment. Wherever applicable, we have explicitly used the term “C₅₅-P” or “lipid substrate” instead of “substrate”, to ensure clarity.

REVIEWER COMMENTS

Reviewer #3 (Remarks to the Author):

In this revision, the authors provide additional experiments, such as a titration of C55-P with respect to *MraY*, and provide a revised conclusion that clarifies a structural, rather than catalytic role, for the C55-P at the dimer interface. Notably, the authors provide experimental evidence that *MraY*, *MurG*, and *MurJ* have increased affinities toward their respective lipid substrates. While the revised manuscript is significantly improved, I still have several serious concerns regarding the MD simulation results and the conclusion drawn from their native MS and MD experiments.

1) The putative C55-P site from the MD simulation.

Based on the MD simulation, the authors conclude that C55-P binds to the dimer interface, where it is surrounded by R340, W253, F254, and Q260. Upon mutation of these residues the protein becomes predominantly monomeric and contains significantly reduced levels of C55-P. The authors conclude that C55-P stabilizes dimerization; therefore, disrupting the interaction between C55-P and *MraY* destabilizes the dimer.

However, I realized that the aforementioned residues form a tight interaction network, which appears to be the major determinant for *MraY* dimerization. In the crystal structure, for example, R340 and W253 form a cation-pi interaction, Q260 and R340 form a H-bond and F254 and W253 form a pi-pi interaction. Therefore, the observed mutational effects on monomer-dimer equilibrium are mainly due to disruption of the dimer interface directly, rather than destabilizing C55-P binding. Primarily, the MD simulation suggests that the phosphate moiety of C55-P appears to be “inserted” into the interaction network adjacent R240, W253, Q260, and F254 (difficult to tell without directly viewing the coordinates), which is a bit unusual. The authors also mention that the crystal structure of *MraY* contains an unresolved electron density profile at the dimer interface (reference #17). Having checked this myself, the density is located inside of the dimer interface, not at the peripheral side of the dimer interface as predicted by MD studies. I could not see any electron density similar to C55-P around the putative binding site predicted by the MD studies. Therefore, it is possible that the experimental design is flawed. Have the authors explored all possible C55-P binding configurations, including the cavity within the dimer interface as well as at the periplasmic interface?

2) The putative Lipid I site from the MD simulation.

Although the authors have changed their view that the C55-P at the dimer interface stabilizes the dimer, it appears that the authors still maintain or infer that the enzymatic activity is at the dimer interface. In Fig.3a and Extended Data Fig. 4C, that authors show that Lipid I binds to the dimer

interface preceding and following the enzyme-catalyzed reaction. It is difficult to understand why Lipid I, the product of the reaction, binds similarly like C55-P to the dimer interface away from the active site, thus competing with the dimer-stabilizing C55-P. Does this imply that newly synthesized Lipid I travels from the active site to the dimer interface and displaces C55-P? If the authors want to maintain this view, they should perform a series of experiments demonstrating when Lipid I and C55-P bind to the dimer interface during the catalytic cycle. In its current standing, the only data indicating Lipid I binding to the dimer interface comes from the MD simulation. However, the authors do not provide control MD simulations to show whether substrates (UDP-MurNAc-pentapeptide, C55-P, and Mg²⁺), in addition to the product (Lipid I), can bind to the active site.

When the MD results are unexpected, experiments should be performed to substantiate the MD results. In these studies, without separate experiments to substantiate the MD results, all native MS data were interpreted based on the MD prediction, which is why many of the previous and some of the current claims do not make sense and against the previous biochemical and structural studies. If the authors want to maintain the views, the authors need to perform a series of biochemical and structural studies to substantiate their claims (Lipid I binding at the dimer interface, C55-P disrupt the existing dimer interface then stabilize the MraY dimer). If not, the authors can remove the MD results for C55-P and Lipid I, and rewrite the manuscript. The newly performed MD studies on MurG look fine to me, as Lipid I and Lipid II bind to the active site.

Reviewer #4 (Remarks to the Author):

I was requested to review the author response to the second reviewer's critical comments on the CG simulations as well as the simulation-related parts of the manuscript.

From the revised manuscript and the response letter, the authors have strongly improved the simulation setup and analysis. The lipid compositions are now more realistic, and the correct lipids species have been simulated, no "proxies". The CG stability of the complexes in CG simulations are now also validated with atomistic MD. I also acknowledge the parametrization of Lipid I and II mentioned in Extended Data Figure 9. The new data support the findings of the manuscript.

However, the documentation of the force fields and simulation systems is incomplete:

1) The CG parametrization strategy and the parameters are not adequately described. The Supplementary information (SI) merely says

"C55-P parameters were as per (ref.)9, and lipid I and II were produced for this study based on C55-PP (ref.9) and parameters for peptidoglycan.[10] See Extended Data Fig. 9e for details of the parameters."

However, I did not find any force field parameters in Ref. 9 (Bolla et al. Angewandte 2020), while C55-P was not even mentioned in Ref. 9. Where did the parameters come from? In addition, the legend of Extended Data Fig. 9 provides no details on the parametrization strategy. What is meant with "were parameterized according to atomistic data" - which atomistic data? From Charmm36 lipids? How were the angles shown in Ext Data Fig. 9e obtained?

2) For the derived force field parameters of Lipid I and II, the authors refer to <https://osf.io/yfstw/>. I could not access this repository anonymously, even after registering at osf.io. Instead, I was promoted to request access and thereby disclose myself. I request that the parameters are freely accessible without restrictions.

3) For the sake of FAIR data and reproducibility, the simulations systems, topologies, and MD parameter (mdp) files should be freely available without restrictions. The authors could use repositories such as Figshare or Zenodo for this purpose. These also allow temporary anonymous access with a secret link, which can be shared with the reviewers during the review process.

Minor:

662 "The sites with the highest affinity re shown in blue" -> are shown in blue

REVIEWER COMMENTS

Reviewer #3 (Remarks to the Author):

In this revision, the authors provide additional experiments, such as a titration of C55-P with respect to MraY, and provide a revised conclusion that clarifies a structural, rather than catalytic role, for the C55-P at the dimer interface. Notably, the authors provide experimental evidence that MraY, MurG, and MurJ have increased affinities toward their respective lipid substrates. While the revised manuscript is significantly improved, I still have several serious concerns regarding the MD simulation results and the conclusion drawn from their native MS and MD experiments.

Our response: We thank the reviewer for their positive comments and careful reading of our manuscript. All the concerns raised have now been fully addressed with new experimental and computational data. Point-by-point responses are described below.

1) The putative C55-P site from the MD simulation.

Based on the MD simulation, the authors conclude that C55-P binds to the dimer interface, where it is surrounded by R340, W253, F254, and Q260. Upon mutation of these residues the protein becomes predominantly monomeric and contains significantly reduced levels of C55-P. The authors conclude that C55-P stabilizes dimerization; therefore, disrupting the interaction between C55-P and MraY destabilizes the dimer.

However, I realized that the aforementioned residues form a tight interaction network, which appears to be the major determinant for MraY dimerization. In the crystal structure, for example, R340 and W253 form a cation-pi interaction, Q260 and R340 form a H-bond and F254 and W253 form a pi-pi interaction. Therefore, the observed mutational effects on monomer-dimer equilibrium are mainly due to disruption of the dimer interface directly, rather than destabilizing C55-P binding.

Primarily, the MD simulation suggests that the phosphate moiety of C55-P appears to be “inserted” into the interaction network adjacent R240, W253, Q260, and F254 (difficult to tell without directly viewing the coordinates), which is a bit unusual. The authors also mention that the crystal structure of MraY contains an unresolved electron density profile at the dimer interface (reference #17). Having checked this myself, the density is located inside of the dimer interface, not at the peripheral side of the dimer interface as predicted by MD studies. I could not see any electron density similar to C55-P around the putative binding site predicted by the MD studies. Therefore, it is possible that the experimental design is flawed. Have the authors explored all possible C55-P binding configurations, including the cavity within the dimer interface as well as at the periplasmic interface?

Our response: The two main issues of concerns raised here are fully addressed in the revised manuscript (highlighted in blue) and below:

A. **Was the MraY-C₅₅-P interaction disrupted in the mutants, or merely the MraY-MraY interactions?**

The reviewer is correct in that the interfacial residues on MraY that coordinate C₅₅-P are involved in dimer stability in the absence of a ligand. We are unable to completely rule out this contribution. We have therefore reflected this possibility in our summary statement on page 9 as follows:

“Since the C₅₅-P coordinating residues at the interfacial sites may also be involved in stabilising the ligand-free form of MraY dimer,¹⁷ there is a possibility that disruption of the dimer interface results in the formation of the monomeric mutants. Nevertheless,

the fact that little or no C₅₅-P/lipid I copurifies with the monomeric mutants indicates a substantial loss of the interfacial binding sites. Overall therefore these results show that disruption of interfacial C₅₅-P binding residues is coupled with destabilization of the MraY dimer.”

However, we would like to highlight two of our experimental observations in Fig 2a,b.

- (i) The point mutants Q260A, K336A and R340A retain their ability to form MraY dimers, yet little or no C₅₅-P is detected copurifying with these mutants.
- (ii) Significant differences in the C₅₅-P binding intensity and stoichiometry are observed between wild-type MraY and Q260A/R340A mutant (Extended Data Fig. 6) when titrated with C₅₅-P. This can only be explained in terms of binding site disruption as both protein variants are compared in their monomeric forms.

Both observations strongly support disruption of C₅₅-P binding at the interfacial sites in the selected mutants. Overall our conclusion, in the previous and current version of the manuscript, is that both dimerization and the interfacial binding of ligands is linked.

We further clarify the above points on page 8-9 as follows:

“Native MS analyses showed that the point mutant Q260A MraY (Fig. 2a) exhibited monomer and dimer populations similar to the wild type protein. However dimers of Q260A MraY were only observed in complex with endogenous lipid I. Little or no protein-bound endogenous C₅₅-P was detected, suggesting selective destabilization of C₅₅-P binding at the interfacial site. Moreover, for the Q260A mutant, each MraY dimer was detected as a complex with 1-4 molecules of endogenous lipid I, indicating that the ligands can bind to MraY at other sites besides the canonical catalytic sites.”

B. Can we attribute unresolved density at the MraY dimer interface to copurified C₅₅-P/lipid I?

The reviewer has a valid point regarding the uncertainties of unresolved density at the dimer interface in the crystal structure of MraY. Can we be completely sure that they cannot correspond to C₅₅-P? Owing to this uncertainty, we have removed the suggestion that the unresolved densities are interfacial C₅₅-P from the discussions on page 19.

Nevertheless, negatively-charged phospholipids have previously been shown to enhance catalytic activities as well as dimerization of MraY (ref. 24). Therefore, we have evaluated the dimer-stabilizing effects of an additional phospholipid (POPG) versus C₅₅-P. We find that the dimer-stabilizing effect of C₅₅-P is much higher than that of the anionic phospholipid POPG. This new data is summarized in Fig. 1c and Extended Data Fig. 3. Described on page 6:

“Anionic phospholipids such as phosphatidyl glycerol have been proposed to stabilise the MraY dimer²⁴. We find however that the dimer-stabilising effect of C₅₅-P is considerably higher than that of 1-palmitoyl-2-oleoyl-*sn*-glycero-3-phospho-(1'-rac-glycerol) (POPG) under the same conditions (Fig.1c, Extended Data Fig. 3).”

Additionally, to provide further evidence in support for a C₅₅-P predicted by MD as capable of interacting on the MraY dimer interface, we have run MD simulations of

monomeric *MraY*. Indeed, we see C_{55} -P binding at R340 on an *MraY* protomer, although this is appreciably lower than in the dimer. These suggested that the interfacial sites for C_{55} -P are more favoured when we have two *MraY* protomers as in a dimer. We have added these observations to the text on page 7:

“Simulating C_{55} -P with an *MraY* protomer (see Methods) further indicates that C_{55} -P can bind to *MraY* via Arg-340, however, with appreciably lower affinity ($k_{off} = 2.1 \pm 0.3 \mu s^{-1}$) than when at the interfacial sites.”

As suggested by the reviewer, we have explored and provided additional data showing that C_{55} -P can bind at multiple sites. These are reported in the new Extended Data Fig. 4. To reflect this, on page 6 we have included:

“Putative binding sites are seen on both periplasmic and cytoplasmic faces of the *MraY* dimer, but the most favourable interactions are predicted to be cytoplasmic, which are more likely to be physiologically relevant (Extended data Fig. 4, Extended data Fig. 5).”

2) The putative Lipid I site from the MD simulation.

Although the authors have changed their view that the C_{55} -P at the dimer interface stabilizes the dimer, it appears that the authors still maintain or infer that the enzymatic activity is at the dimer interface. In Fig.3a and Extended Data Fig. 4C, that authors show that Lipid I binds to the dimer interface preceding and following the enzyme-catalyzed reaction. It is difficult to understand why Lipid I, the product of the reaction, binds similarly like C_{55} -P to the dimer interface away from the active site, thus competing with the dimer-stabilizing C_{55} -P. Does this imply that newly synthesized Lipid I travels from the active site to the dimer interface and displaces C_{55} -P? If the authors want to maintain this view, they should perform a series of experiments demonstrating when Lipid I and C_{55} -P bind to the dimer interface during the catalytic cycle. In its current standing, the only data indicating Lipid I binding to the dimer interface comes from the MD simulation. However, the authors do not provide control MD simulations to show whether substrates (UDP-MurNAC-pentapeptide, C_{55} -P, and Mg^{2+}), in addition to the product (Lipid I), can bind to the active site. When the MD results are unexpected, experiments should be performed to substantiate the MD results. In these studies, without separate experiments to substantiate the MD results, all native MS data were interpreted based on the MD prediction, which is why many of the previous and some of the current claims do not make sense and against the previous biochemical and structural studies. If the authors want to maintain the views, the authors need to perform a series of biochemical and structural studies to substantiate their claims (Lipid I binding at the dimer interface, C_{55} -P disrupt the existing dimer interface then stabilize the *MraY* dimer). If not, the authors can remove the MD results for C_{55} -P and Lipid I, and rewrite the manuscript. The newly performed MD studies on MurG look fine to me, as Lipid I and Lipid II bind to the active site.

Our response: We thank the reviewer for their comments. However, we would like to respectfully point out that our results and conclusions do not contradict those from the previous assays in the literature. First, we do not claim that the catalysis takes place at the dimer interface (for example, please see Fig. 3 legend). Secondly, our MD data do not claim that the dimer interface is the only place where C_{55} -P or lipid I can bind to *MraY*. It is common for membrane proteins to bind lipids for structural purposes, for example, *Nature* **510**, 172-175 (2014), *Nature* **541**, 421-424 (2017), *Cell Chem. Biol.* **25**, 840-848 e4 (2018). In fact, previous reports indicated that the binding of negatively charged lipids to *E. coli* *MraY* is required for dimerization and full activity (*Elife* **6**, e20954. (2017)). In our study, *MraY* copurified with up to 4 endogenous ligands per *MraY* dimer (e.g., the Q260A mutant, Fig 2a, see RF Fig1 below). Despite multiple detergent washes during the purification steps, which largely removed all other phospholipids, lipid I remains.

RF Fig.1: A deconvoluted spectrum of Q260A MraY mutant. Shown are the peaks corresponding to MraY monomer, and the dimer captured in complex with 1-4 endogenous lipid I molecules.

Since each MraY protomer carries no more than a single catalytic site, there must be other sites enabling binding of additional lipid I molecules. Experimental identification of specific residues on MraY interacting with *t* copurified lipid I is beyond the scope of the present study. We have however made several revisions to the relevant texts as follows:

Page 6:

“Putative binding sites are seen on both periplasmic and cytoplasmic faces of the MraY dimer, but the most favourable interactions are predicted to be cytoplasmic, since they more likely to be physiologically relevant (Extended data Fig. 4, Extended data Fig. 5).”

We describe the MD results by including on page 7:

When MraY was simulated with lipid I alone, we found that lipid I interacts considerably at the interfacial and active site regions (Extended Data Fig. 5a,b). Lipid I also binds to other positions around MraY, albeit with lower affinity (Extended Data Fig. 5c, f). Interestingly, simulations of the MraY dimer with both C_{55} -P and lipid I, present in the model membrane, show that lipid I occupies the catalytic sites for higher percentage of the simulation time than C_{55} -P (Supplementary Table S3). Analogously C_{55} -P occupies the interfacial sites for more of the simulation time than lipid I (Supplementary Table S3). Overall, these data indicate that MraY can interact with both C_{55} -P and lipid I at multiple sites, including around the dimer interface, and suggests a putative role of C_{55} -P and lipid I in conferring stability to the MraY dimer.

Guided by the interfacial C_{55} -P binding sites predicted by the MD simulation results, we generated several mutants of MraY to probe experimentally the link between substrate binding and dimerization. Using identical expression and purification conditions employed for wild-type protein we examined K336A, Q260A, R340A, Q260A/R340A, and W253A/F254A/R340A mutants of MraY. Native MS analyses showed that the point mutants Q260A and K336A MraY exhibited monomer and dimer properties similar to the wild type (Fig.2b). However, dimers of both proteins were observed only in complex with endogenous lipid I, little or no protein-bound endogenous C_{55} -P was detected. For the Q260A mutant, each MraY dimer was detected in complex with between 1-4 molecules of endogenous lipid I, indicating that the ligands can bind to MraY at other sites besides the canonical catalytic sites.

We have added to the figure legends for Extended Data Fig. 4 and 5:

Page 30:

Extended Data Fig. 4. MD simulations of *MraY* and C_{55} -P. a-d. Representative pose of C_{55} -P bound to *MraY* from atomistic molecular dynamic simulations. Panels **a** shows C_{55} -P binding to the dimer interface, **b** to the active site, while **c** and **d** show a selected lower affinity poses around the *MraY* dimer. The protein backbone chain is shown as a cartoon, with one protomer coloured gold and the other cyan. A C_{55} -P molecule is shown as coloured spheres (hydrophobic tail, green).

Page 31:

Extended Data Fig. 5. Poses of lipid I around *MraY*. a-d. A number of lipid I binding poses on the *MraY* dimer generated for lipid I from the simulation data using PyLipID²⁸, and converted to atomic detail for visualisation using CG2AT.³⁹ Panels: **a** shows binding to the dimer interface, **b** to the active site, while **c** and **d** show a select few of alternative lower affinity binding poses of lipid I around the *MraY* dimer.

The reviewer's speculation that " C_{55} -P disrupts the existing dimer interface rather than stabilizing the *MraY* dimer" seems unlikely. We predict that C_{55} -P or lipid I should be able to bind the *MraY* dimer interface without disrupting the dimer. Our experimental data clearly show that the *MraY* dimer can exist in a ligand free state, while the binding of C_{55} -P, lipid I or both can further enhance the dimer population (e.g., Fig 1a). Since lipid I can bind at other sites beside the catalytic site or dimer interface, we have redrawn the schematic depiction of ligand-bound dimer throughout to reflect our contention that binding is not exclusive to the catalytic sites.

To improve on the rigour of our MD simulations and to help give insight into the competition between C_{55} -P and lipid I at the *MraY* sites, we have now performed extensive rounds of simulations of *MraY* dimer in the presence of both C_{55} -P and lipid I. The new data, now reported is Supplementary Table S3, shows that binding of C_{55} -P is more favourable at the interfacial site. By contrast lipid I binds more favourably near the catalytic sites, which directly addresses the reviewer's concerns.

Accordingly, we have revised the methods for the MD simulations in the supplementary information on page 8-9, including the following:

"For *MraY*, additional systems were built with 62% POPE, 23% POPG, 10% cardiolipin and 2.5% of both C_{55} -P and lipid I.

C_{55} -P parameters were as per previous reports⁹ (where it is termed UDP), and parameters for lipid I and II were produced for this study, see Extended Data Fig. 9e for mapping details. The tails of these were based on C_{55} -PP from ref.⁹, for which the bonded terms were based on atomistic MD of C_{55} -PP in a POPE membrane. The lipid I and II headgroup parameters were taken from peptidoglycan.¹⁰ To finalise the molecules, additional angle terms were applied between the head group and tails, as per Extended Data Fig. 9e. These were based upon atomistic MD simulation (see below). Virtual CG beads were imposed on the atomistic system by clustering groups of atoms according to our mapping scheme. Angles between "beads" were then computed using gmx gangle. These values were used to define the lipid II CG parameters, with force constants set to ensure the CG and AT distributions matched (Extended Data Fig. 9e)."

Reviewer #4 (Remarks to the Author):

I was requested to review the author response to the second reviewer's critical comments on the CG simulations as well as the simulation-related parts of the manuscript.

From the revised manuscript and the response letter, the authors have strongly improved the simulation setup and analysis. The lipid compositions are now more realistic, and the correct lipids species have been simulated, no "proxies". The CG stability of the complexes in CG simulations are now also validated with atomistic MD. I also acknowledge the parametrization of Lipid I and II mentioned in Extended Data Figure 9. The new data support the findings of the manuscript. However, the documentation of the force fields and simulation systems is incomplete:

Our response: We thank the reviewer for their careful reading of the manuscript and their kind remarks. We have taken on board all identified concerns and have ensured complete documentation of our MD simulation methods.

1) The CG parametrization strategy and the parameters are not adequately described. The Supplementary information (SI) merely says

"C55-P parameters were as per (ref.)⁹, and lipid I and II were produced for this study based on C55-PP (ref.⁹) and parameters for peptidoglycan.^[10] See Extended Data Fig. 9e for details of the parameters."

However, I did not find any force field parameters in Ref. 9 (Bolla et al. *Angewandte* 2020), while C55-P was not even mentioned in Ref. 9. Where did the parameters come from? In addition, the legend of Extended Data Fig. 9 provides no details on the parametrization strategy. What is meant with "were parameterized according to atomistic data" - which atomistic data? From Charmm36 lipids? How were the angles shown in Ext Data Fig. 9e obtained?

Our response: We have now improved the description of the parameterization process. In particular, we have details of the methods describing how the lipid II atomistic simulations were run, and a new section that outline the steps and process used to use the atomistic data to refine the CG parameters. We have also highlighted that in reference 9, C55-P was instead termed UDP (undecaprenyl-pyrophosphate) as follows (page 8):

"C₅₅-P parameters were as per previous reports⁹ (where it is termed UDP), and parameters for lipid I and II were produced for this study, see Extended Data Fig. 9e for mapping details. The tails of these were based on C₅₅-PP from ref.⁹, for which the bonded terms were based on atomistic MD of C₅₅-PP in a POPE membrane. The lipid I and II headgroup parameters were taken from peptidoglycan.¹⁰ To finalise the molecules, additional angle terms were applied between the head group and tails, as per Extended Data Fig. 9e. These were based upon atomistic MD simulation (see below). Virtual CG beads were imposed on the atomistic system by clustering groups of atoms according to our mapping scheme. Angles between "beads" were then computed using gmx gangle. These values were used to define the lipid II CG parameters, with force constants set to ensure the CG and AT distributions matched (Extended Data Fig. 9e)."

2) For the derived force field parameters of Lipid I and II, the authors refer to <https://osf.io/yfstw/>. I could not access this repository anonymously, even after registering at osf.io. Instead, I was promoted to request access and thereby disclose myself. I request that the parameters are freely accessibly without restrictions.

Our response: We thank the reviewer for this feedback. We typically make the OSF account public immediately after a manuscript is accepted, or upon request prior. We agree with the reviewer that these might be of use for the reviewing process, so we have now made the OSF public ahead of reviewing.

3) For the sake of FAIR data and reproducibility, the simulations systems, topologies, and MD parameter (mdp) files should be freely available without restrictions. The authors could use repositories such as Figshare or Zenodo for this purpose. These also allow temporary anonymous access with a secret link, which can be shared with the reviewers during the review process.

Our response: We appreciate this concern raised by the reviewer. However, topologies and mdps are not typically made public until a manuscript is accepted for publication. We agree with the reviewer for their interest the mdps have now been added to the OSF, to accompany the existing parameter files. These should now be accessible without divulging any identifying details. The other topologies used were standard for Martini 2.2.

Minor:

662 "The sites with the highest affinity re shown in blue" -> are shown in blue.

Our response: We apologize for the typo. This sentence is now corrected as follows:
"The equivalent sites on each side of the MraY dimer **are** shown in the same colour. The sites with the highest affinity are shown in blue with the principal contributing residues shown as coloured spheres."

REVIEWERS' COMMENTS

Reviewer #3 (Remarks to the Author):

This revised manuscript is substantially improved from the previous version and has addressed my concerns. I recommend its publication to Nature Communications.

Reviewer #4 (Remarks to the Author):

My concerns on the MD simulation results and methods have been mostly adequately addressed.

I request the following additions for re-use of the force field parameters:

1) Thank you for giving me access to <https://osf.io/yfstw/>. I noticed that force field files (itp files) are available, but no structure files (pdb or gro). Without structures, the itp files are hardly useful. Please add a gro file for each lipid to <https://osf.io/yfstw/>.

2) As far as I can see, <https://osf.io/yfstw/> does not have a permanent DOI, so the authors may consider providing the itp, gro and mdp files as a supporting material zip folder with the publication. Since these files are small, adding them to the SI should be no problem.

2) Regarding simulation systems (my previous point 3), I still request that these are made freely available via some database, of course *after* acceptance of the article. During the review process, however, simulation systems can be made available to reviewers via a secret link. Zenodo supports this feature, probably also Figshare. From my side, it is sufficient if the Nature Communications editors oversee the publication of the simulation systems before the formal acceptance of the manuscript. But I can offer to look at the database and report whether the simulation systems are complete.

REVIEWER COMMENTS

Reviewer #3 (Remarks to the Author):

This revised manuscript is substantially improved from the previous version and has addressed my concerns. I recommend its publication to Nature Communications.

Our response: We appreciate the reviewer for their careful reading of the manuscript and for positive recommendation.

Reviewer #4 (Remarks to the Author):

My concerns on the MD simulation results and methods have been mostly adequately addressed.

Our response: We thank the reviewer for their constructive comments.

I request the following additions for re-use of the force field parameters:

1) Thank you for giving me access to <https://osf.io/yfstw/>. I noticed that force field files (itp files) are available, but no structure files (pdb or gro). Without structures, the itp files are hardly useful. Please add a gro file for each lipid to <https://osf.io/yfstw/>.

Our response: As requested, we have now deposited a zip folder (named gros.zip) containing the gro files for each lipids to have been deposited and publicly released in figshare via <https://doi.org/10.6084/m9.figshare.19403852.v2>.

2) As far as I can see, <https://osf.io/yfstw/> does not have a permanent DOI, so the authors may consider providing the itp, gro and mdp files as a supporting material zip folder with the publication. Since these files are small, adding them to the SI should be no problem.

Our response: We have deposited all the relevant files at figshare which has a permanent DOI.

2) Regarding simulations systems (my previous point 3), I still request that these are made freely available via some database, of course *after* acceptance of the article. During the review process, however, simulation systems can be made available to reviewers via a secret link. Zenodo supports this feature, probably also Figshare. From my side, it is sufficient if the Nature Communications editors oversee the publication of the simulation systems before the formal acceptance of the manuscript. But I can offer to look at the database and report whether the simulation systems are complete.

Our response: As requested, we have included an `example_input_coordinate.zip` file to figshare and can be publicly accessed at <https://doi.org/10.6084/m9.figshare.19403852.v2>.